# Has erosion globally increased? Long-term erosion rates as a function of climate derived from the impact crater inventory

**Stefan Hergarten**[1] **and Thomas Kenkmann**[1]

[1]Institut für Geo- und Umweltnaturwissenschaften, Albert-Ludwigs-Universität Freiburg, Freiburg i. Br., Germany

**Correspondence:** S. Hergarten
(stefan.hergarten@geologie.uni-freiburg.de)

**Abstract.** Worldwide erosion rates seem to have increased strongly since the beginning of the Quaternary, but there is still discussion about the role of glaciation as a potential driver and even whether the increase is real at all or an artefact due to losses in the long-term sedimentary record. In this study we derive estimates of average erosion rates on the time scale of some tens of million years from the terrestrial impact crater inventory. This approach is completely independent from all other methods to infer erosion rates such as river loads, preserved sediments, cosmogenic nuclides and thermochronometry. Our approach yields average erosion rates as a function of present-day topography and climate. The results confirm that topography accounts for the main part of the huge variation of erosion on Earth, but also identifies a significant systematic dependence on climate in contrast to several previous studies. We found a fivefold increase in erosional efficacy from the cold regimes to the tropical zone and that temperate and arid climates are very similar in this context. Combining our results to a worldwide mean erosion rate, we found that erosion rates on the time scale of some tens of million years are at least as high as present-day rates and suggest that glaciation has a rather regional effect with a limited impact at the continental scale.

## 1 Introduction

The origin of the apparently huge increase of worldwide erosion in the late Cenozoic era is one of the major puzzles in the younger geologic history of our planet (Molnar and England, 1990; Zhang et al., 2001; Molnar, 2004; Willenbring and von Blanckenburg, 2010; Herman et al., 2013; Wang et al., 2014; Marshall et al., 2015; Willenbring and Jerolmack, 2015). As high temperatures facilitate weathering of rocks, the cooling climate during the Cenozoic era should rather result in decreasing erosion rates, bringing Pleistocene glaciation as a major driver of erosion into discussion (Yanites and Ehlers, 2012; Brocklehurst, 2013; Egholm, 2013; Pedersen and Egholm, 2013; Koppes et al., 2015; Herman and Champagnac, 2016).

However, most of the knowledge about the apparent worldwide increase relies on estimates of long-term erosion rates from preserved sediments in the oceans (e.g., Wilkinson and McElroy, 2007). Based on Sadler's theory (Sadler, 1981) addressing the scale dependence of sedimentary records, the existence of a worldwide increase has already been questioned by Willenbring and von Blanckenburg (2010). In their study, the theoretical arguments were supported by Beryllium isotope ratios revealing no systematic variation in the overall sediment delivery rates during the last 12 Ma. On the other hand, a recent study on thermochronometric data not depending on the long-term sedimentary record has revealed a strong increase at least in some mountainous regions with high erosion rates during the last 10 Ma (Herman et al., 2013). However, potential systematic errors in thermochronometry have been discussed in the previous years (Valla et al., 2010; Willenbring and Jerolmack, 2015), and the worldwide increase found by Herman et al. (2013) has recently been questioned by Schildgen et al. (2018).

Worldwide present-day erosion rates have also been addressed in several studies. However, all approaches suffer

from the need to upscale point data, leading to a large variation in the estimates of the worldwide mean rate (see, e.g., the compilation by Willenbring et al., 2013). As an additional source of uncertainty, an increasing portion of the eroded sediments is trapped in artificial reservoirs today (Syvitski et al., 2005).

As topography, climate, and lithology are the main controls on erosion, there have been several approaches to quantify the contribution of these components. Concerning the variation over Earth's surface, topography has the strongest influence. Several metrics have been introduced in order to relate topography to erosion rates at different scales. The seminal study of Ahnert (1970) suggested a linear dependency of the erosion rate on mean relief (difference between maximum and minimum elevation) even without any correlation to precipitation. Later studies used either relief, slope or modal elevation and also obtained a linear or almost linear increase of the erosion rate with the respective geomorphic property (for a comparison see Summerfield and Hulton, 1994). In the context of active tectonics, channel slopes and channel steepness have been widely used (e.g., Wobus et al., 2006).

Although the effect of climate on erosion has been addressed in several publications at least indirectly (see references above), the number of studies finally arriving at a clear relationship between long-term erosion and climate seems to be limited. In a study on organic carbon fluxes, Ludwig and Probst (1996) also estimated sediment fluxes into the oceans and found a strong correlation with climate. According to their results, the wet tropic climate zone contributes about 44 % to the worldwide sediment supply, while the tundra and taiga zone contributes only 5 %, although both cover the same area on Earth in total. In contrast, the presumably most comprehensive compilation of millennial-scale erosion rates (Portenga and Bierman, 2011) involving cosmogenic nuclide data from almost 1600 drainage basins and outcrops even yielded an unsystematic dependence on climate, presumably because the dominant effect of topography shadows all other influences. A weak effect of climate was also found by Riebe et al. (2001) even at smaller scales. In turn, recent studies (Moon et al., 2011; Ferrier et al., 2013) have at least confirmed the correlation between precipitation and erosion rates that is implicitly assumed in all models of fluvial erosion within regions with high contrasts in precipitation.

## 2   Deriving erosion rates from the impact crater inventory

In planetary geology, the inventory of impact craters provides the most valuable data for unraveling the geological history (e.g., Neukum et al., 2001; Stöffler and Ryder, 2001). The terrestrial inventory, however, has not been exploited systematically beyond the research on impact processes themselves, probably due to its small extent compared to other planets

and to its uncertain completeness. However, recent studies suggest that the terrestrial crater record is by far not as incomplete as it was presumably assumed in the past. Taking into account the age distribution of the Earth's crust, it was found that the inventory of the craters at least 85 km wide may already be complete (Johnson and Bowling, 2014). A subsequent study (Hergarten and Kenkmann, 2015) also considering the consumption of craters by erosion even revealed no evidence for any incompleteness in the crater record above 6 km diameter exposed at the ice-free part of Earth's land surface. This study also quantified the potential incompleteness in the diameter range from 0.25 km to 6 km.

Erosion of craters was described by a simple model that can be visualized as a rasp in this approach. It was assumed that the region around a crater is uniformly eroded at a given constant rate, and that the crater remains detectable until the total erosion reaches a given depth. As the impact origin of crater-like topographic features is unequivocally proven by the existence of rocks altered by the impact process, this depth was assumed to be the greatest depth where these shock effects typically occur. Relating this depth to the crater diameter provided an estimate of the lifetime of craters as a function of the diameter and the erosion rate.

Using the presumably best estimate of the terrestrial crater production rate available (Bland and Artemieva, 2006), it was found that the expected density $N$ (number per area) of craters with a diameter of at least 0.25 km at an erosion rate $r$ is

$$N = \frac{I}{r} \tag{1}$$

with a constant $I = 4.94 \times 10^{-5} \; \frac{\mathrm{m}}{\mathrm{Ma\,km^2}}$ (Hergarten and Kenkmann, 2015, Eq. 9). The value of $I$ takes into account the crater production rate, the depth-diameter relation of craters and an estimate of the potential incompleteness of the inventory in the diameter range from 0.25 km to 6 km.

If erosion is spatially homogeneous in the considered domain, Eq. (1) immediately predicts the expected number $n$ of craters as

$$n = AN = \frac{AI}{r} \tag{2}$$

where $A$ is the size of the domain. For heterogeneous erosion, Eq. (1) yields

$$n = \int N dA = I \int \frac{1}{r} dA. \tag{3}$$

Inverting this relationship allows for an estimation of some spatially and temporally averaged erosion rate from the number of impact craters in a given region.

At this point it is noteworthy that the spatial average is not an average over the locations of the existing craters, but over the entire area. In other words, the approach does not only derive information on erosion rates from regions where craters

are, but also from crater-free regions. It therefore avoids the potential sampling bias due to an uneven distribution of locations that might occur in all methods where erosion rates measured at points or over small areas must be transferred to large areas.

In turn, the occurrence of $r$ in the denominator in Eq. (3) reveals that the number of craters in a given region does not yield the arithmetic mean erosion rate (as it is relevant, e.g., for the sediment yield), but the harmonic mean rate. The latter is always lower than the arithmetic mean, and the discrepancy increases with increasing spatial heterogeneity. Let us illustrate the difference by a simple example (which will be revisited in Sect. 8.5). If the entire surface of Earth consisted of two parts of equal sizes where one part has a high erosion rate of $r_h = 120$ m/Ma and the other a low erosion rate $r_l = 30$ m/Ma, the arithmetic mean rate would be 75 m/Ma. The harmonic mean erosion rate would, however, be only $\left(\frac{1}{2}\left(r_h^{-1} + r_l^{-1}\right)\right)^{-1} = 48$ m/Ma and thus be more than one third lower than the arithmetic mean rate. Taking this discrepancy into account, it can be expected that the harmonic mean rate for the entire ice-free land surface of $r = 59$ m/Ma obtained by Hergarten and Kenkmann (2015) significantly underestimates the arithmetic worldwide mean.

Overcoming this limitation is one of the main goals of this paper. Subdividing the total surface into a sufficient number of domains and then averaging over these domains seems to be a straightforward idea, but is limited by the low number of impact craters exposed at Earth's surface. At the time of the original study, the Earth Impact Database (http://www.passc.net/EarthImpactDatabase/) comprised 188 terrestrial craters in total with only 112 of them exposed at the surface and wider than 0.25 km. While two more craters have been added to the database until now, the number of relevant craters is still 112. While this number in total provides a moderate statistical error of about 10 % (standard deviation of Poisson distribution), the statistical errors rapidly increase if the number of craters per domain decreases. In particular, crater-free domains would cause serious problems as the estimated erosion rate would be infinite (with an infinite error, too). Therefore, a more sophisticated approach is required; it will be explained in the following sections.

The original estimate of $r = 59$ m/Ma contains a second source of potentially large systematic errors. Craters are not only consumed by erosion, but may also be buried by sediments. This process would erroneously be interpreted as erosion in the model. As craters may form local sinks for sediments, the local sediment accumulation rates in a crater may be much higher than regional erosion or sedimentation rates, so that craters may become invisible by burial even more rapidly than by erosion. Thus, sediment deposition in parts of the considered domain leads to an overestimation of the erosion rate. So the original estimate contains two sources of systematic errors in opposite directions.

## 3 The influence of topography on erosion

Topography contributes the largest part to the spatial variation in erosion, and several metrics of topography were proposed in order to quantify the dependence of erosion rate on topography (e.g., Summerfield and Hulton, 1994). Almost 50 years ago, Ahnert (1970) found a linear dependency of erosion rate on relief at large scales. Since digital elevation models (DEM) have become widely available, local slope has often been preferred over relief (Montgomery and Brandon, 2002; Whipple et al., 2013; Willenbring et al., 2013), mainly because it allows for a higher spatial resolution. In particular when point data must be transferred to large areas, the better spatial resolution is an advantage. In the field of tectonic geomorphology, preference is given to the analysis of individual channel profiles. In combination with models of fluvial erosion, they can even be used for reconstructing the tectonic history of a given region (e.g., Wobus et al., 2006).

However, the analysis of river profiles requires a careful consideration and appears not yet to be ready for fully automatic application at large scales. Even the analysis of the local slope involves some pitfalls since mean slopes computed from a DEM strongly depend on its resolution (Willenbring et al., 2013, 2014). As the spatial resolution is not important for our application, we return to the old concept of the relief. It is quite robust against the resolution of the DEM and can be computed almost as efficiently as local slope if taken over squares instead of circles (as mostly done). In this study we measure relief over squares of 10 km edge length using the worldwide ETOPO1 DEM with a mesh width of one arc minute and also verified that our results basically persist for squares of 5 km and 20 km edge length as originally used by Ahnert (1970).

In order to verify the relationship between relief and erosion rate on large scales, we first subdivide the ice-free land surface into the six basic types of continental crust (shield, platform, orogen, basin, igneous province, extended crust) defined in the world map of the main geological provinces provided by the USGS (http://earthquake.usgs.gov/data/crust/type.html). Figure 1 relates the mean apparent erosion rates (rates of crater consumption) estimated from Eq. (2) for each of the types of crust to their average 10 km relief. The three crustal types shield, orogen, and igneous province expected to be predominantly erosive regimes differ strongly in their mean relief, but show a strikingly linear relationship between the rate of crater consumption $r$ and the mean relief $\Delta$,

$$r = s\Delta. \tag{4}$$

The three other types, platform, basin, and extended crust, are characterized by much higher rates in relation to their mean relief, suggesting that deposition of sediments significantly contributes to the consumption of craters here. We therefore consider only the three predominantly erosive

crustal types in our analysis and assume a linear relationship between relief and long-term erosion rate.

If we forget for the moment that Eq. (4) is applicable to large scales only, inserting it into Eq. (3) allows for estimating $s$ from the total number of craters $n$ according to

$$s = \frac{I}{n} \int \frac{1}{\Delta} dA. \tag{5}$$

With regard to the applicability of Eq. (4) at large scales only, the integral should be replaced by a discrete sum over a finite set of (sufficiently large) subdomains,

$$s = \frac{I}{n} \sum_{i=1}^{k} \frac{A_i}{\Delta_i}, \tag{6}$$

where $k$ is the number of subdomains, and $A_i$ and $\Delta_i$ are the size and the mean relief of each subdomain, respectively.

As the key point of this consideration, the estimate of $s$ is still obtained from the total number of craters following a Poissonian distribution and does not rely directly on the number of craters in each subdomain. Thus, the parametric relationship between relief and erosion rate allows to take the heterogeneity arising from the strong worldwide variation in relief into account without increasing the statistical errors from the limited number of craters on Earth.

## 4    The influence of climate on erosion

While the strong effect of relief on erosion rates can be taken into account using the parametric approach discussed in the previous section, it is not possible to proceed in this direction for the further factors controlling erosion. This would require a quantitative relationship between erosion rate and any property where data are avaliable at the entire surface (e.g., precipitation), whereas pure correlations do not help. Thus, a further reduction of the underestimation arising from the harmonic mean can only be achieved by a subdivision of Earth's surface into independent domains where the value of $s$ differs among the domains. As the number of craters per domain is lower than the total number, we reduce the systematic error for the price of increasing statistical uncertainty then. Thus, the domains should be chosen in such a way that they capture a major part of the variation of erosion going beyond the effect of topography, but the number should not be too high.

As a tradeoff between the expected effect on erosion and the number of domains we consider the primary classes tropical (A), arid (B), temperate (C), cold (D), and polar tundra (ET) of the Köppen-Geiger classification of the recent climate (Peel et al., 2007) shown in Fig. 2. The class polar frost (EF) was omitted as it primarily consists of ice-covered areas. A value of $s$ is then assigned to each of the climate zones by applying Eq. (6). The result $s$ is a lumped value summarizing all influences on erosion going beyond the topography. As variations in lithology should not be significant at the large scales considered here, $s$ can be seen as a measure for the erosional efficacy of the respective climatic regime, so that we will use this term throughout the paper.

For applying Eq. (6) to each of the climate zones, the respective domain must be further subdivided into subdomains capturing the variation in relief reasonably well. For this we use the six main types of crust mentioned above, where only the three predominantly erosive types are used for estimating erosion rates. Considering unconnected parts of the same type of crust as separate subdomains, this yields a subdivision of the predominantly erosive provinces into 89 subdomains (13–22 per climate zone) with sizes from about $1600\,\mathrm{km}^2$ to about 11 million $\mathrm{km}^2$ (for details see supplementary material).

The resulting erosional efficacies of the climate zones are shown in Fig. 3a. In contrast to some of the previous studies (Ahnert, 1970; Riebe et al., 2001; von Blanckenburg, 2006; Portenga and Bierman, 2011), we found a clear systematic dependence on climate, at least for those classes primarily defined by temperature (A, C, D, ET). While the two cold Köppen-Geiger classes D and ET are very similar ($s = 0.13\,\mathrm{Ma}^{-1}$), the erosional efficacy of the tropical zone ($s = 0.62\,\mathrm{Ma}^{-1}$) is almost 5 times higher. With $s = 0.30\,\mathrm{Ma}^{-1}$, the temperate class is close to the (geometric) mean of the two extremes. This clear trend goes along with the increase in both temperature and precipitation from polar to tropical regions.

The result for the arid zone, $s = 0.30\,\mathrm{Ma}^{-1}$, suggests that that the erosional efficacy of the arid climate is as high as that of temperate climate. This may be surprising as the arid zone is defined by low precipitation in relation to the temperature and covers a wide range of temperatures. However, the major part of the worldwide arid range is characterized by high temperatures (Köppen-Geiger classes BWh and BSh), so that the mean rate of chemical weathering should indeed be high here. But as water is the main agent for mechanical erosion and sediment transport, the result that the high temperatures are able to compensate the low precipitation compared to the temperate climate is still surprising.

In this context, the time scale of the considered mean values must be taken into account, too. Based on the estimated lifetimes of the considered impact craters, a time scale of 10–100 Ma was estimated (Hergarten and Kenkmann, 2015). Mean temperatures have varied over this time span, accompanied by changes in overall precipitation, so that the climate classes primarily defined by temperature have shifted with the coldest and warmest classes extending or shrinking. Furthermore, continents have moved on this time scale. So it should be mentioned that our estimate $s$ is, strictly speaking, not the actual erosional efficacy of the present-day climate, but the erosional efficacy of the part of Earth's surface belonging to the considered climate zone measured over a long time span into the past. As discussed in Sect. 8.5, the real differences in erosional efficacies of the climatic zones may be higher than suggested by our study. This may also apply to

the surprisingly high erosional efficacy of the arid zone. Our results do not refute the importance of water for erosion, but may tentatively suggest that the present-day arid zone may have been wetter than today in the past.

The clear relationship between erosional efficacy and climate (Fig. 3a) is slightly blurred after computing absolute erosion rates using the mean relief (Fig. 3b). The mean relief of the predominantly erosive provinces is highest in the temperate zone, $\Delta > 500\,\mathrm{m}$, while it is lower than 300 m in both the tropical and the arid zone and on an intermediate level ($\Delta \approx 400\,\mathrm{m}$) in the two cold regimes. As a consequence, the variation in the absolute erosion rates shown in Fig. 3b is smaller than the variation in $s$, and the temperate zone is characterized by a high mean erosion rate almost catching up with the tropical zone.

Figure 3c shows the extrapolation of the results for the entire ice-free surface including the types of crust excluded not taken into account so far (platform, basin, and extended crust). For the extrapolation we assumed that parts of these provinces are erosive with the erosional efficacy of the respective climate zone, while the rest is dominated by sediment deposition. Assuming that regions of sedimentation have a very small (strictly speaking, zero) relief, the erosional efficacy is also valid for these mixed zones and thus for the entire climate zone (including the regions of sedimentation). Depending on the climate class, the mean erosion rates decrease by 13 % to 32 % due to the lower mean relief of the extrapolated provinces. However, the results are qualitatively similar to those obtained for the predominantly erosive provinces.

The area-weighted mean over the five climatic zones (Fig. 3c) yields a worldwide mean erosion rate of $r = 78\,\mathrm{m/Ma}$ (107 m/Ma for the predominantly erosive provinces) with 95 % confidence limits of 52 m/Ma and 116 m/Ma (see Appendix A). Our result is almost 40 % higher than the mean Pleistocene (2.58–0.01 Ma b.p.) erosion rates of $r = 56\,\mathrm{m/Ma}$ obtained from preserved sediments (Wilkinson and McElroy, 2007). The latter value is even close to our lower 95 % confidence limit, and all known values for earlier periods of Earth's history are even lower. This result already suggests that erosion rates in the past might be much higher than those obtained from preserved sediments. We will return to this point after considering the time scale addressed by our approach more thoroughly (Sect. 6).

## 5 The spatial distribution of erosion on Earth

Figure 4 shows a world map of the estimated erosion rates using the 10 km relief on a $0.1° \times 0.1°$ lattice and the values $s$ of the respective climate zones. The dominance of topography over climate is immediately visible. While the mean relief amounts to 260 m, the maximum relief is 5887 m, which is more than 20 times larger than the mean relief. In contrast, the erosional efficacy $s$ differs only by about a factor of 5 be-

tween the warmest and the coldest climate classes. However, very high erosion rates above 1000 m/Ma occur over considerable areas only in combination of tropical climate and high relief. The largest domain with estimated erosion rates above 1000 m/Ma is found in New Guinea.

Figure 5 compares the estimated erosion rates with the present-day erosion erosion rates published by Wilkinson and McElroy (2007) based on the study of Ludwig and Probst (1996). As this study focused on organic carbon, specific bioclimatic zones were defined instead of the Köppen-Geiger climate classes used in our study. Therefore a direct comparison based on climate zones is not possible, so that a comparison by latitude remains as the most convenient approach.

In general, our estimates show a much more homogeneous distribution on Earth than the estimates of the recent erosion rates. The quite inhomogeneous distribution of the latter is reflected in a strong asymmetry between the two hemispheres, a strong decrease towards the polar regions and a pronounced peak at $20°N$. However, the smaller variation of our results is not surprising since our results are an average over a long time span where climate has changed and even continents have moved.

As shown in Fig. 6, the contribution of the area with an erosion rate greater than $r$ can be approximated well by an exponential distribution $C(r) = 0.25 \exp(-\frac{r}{200\,\mathrm{m/Ma}})$ at high erosion rates above 250 m/Ma. This means that the area on Earth with an erosion rate greater than $r$ decreases by about 40 % if $r$ increases by 100 m/Ma. Qualitatively the same behavior was found for soil losses at the plot scale, but with a decay constant about 5 times smaller (Wilkinson and McElroy, 2007). Even more striking, there is a significant deviation from the exponential decay at erosion rates below 250 m/Ma. The exponential part covers only 8 % of the total ice-free land surface. The steeper decrease in the cumulative distribution at low erosion rates indicates that these areas contribute much more to the total area than the exponential tail. However, when considering the contribution to the worldwide erosion, a different behavior is observed. Here, the contribution of the large area with small erosion rates is not so high. Using our estimate of the worldwide mean erosion rate of 78 m/Ma, the data reveal that only about 25 % of the total land surface have an erosion rate above the mean, but these 25 % contribute about 75 % to total erosion. This 75 to 25 relation describes a more uneven distribution than Willenbring et al. (2014) obtained (about 70 to 30), but it is less inhomogeneous than the 80 to 20 relation often referred to as Pareto's principle in many contexts.

## 6 The time scale of the terrestrial crater inventory

As the lifetime of a crater at a given erosion rate depends on its size, the number of craters of different sizes should reflect the mean erosion rate over different time intervals. We might therefore think about an inversion approach us-

ing the crater inventory as a function of the crater size for deriving time-resolved erosion rates. Alternatively, we could use the very good fit of the inventory assuming a constant erosion rate obtained by Hergarten and Kenkmann (2015) as evidence for a constant erosion rates over millions of years. However, the spatial variation of erosion rates immediately tears down these such ideas. Mainly due to the variation of relief, erosion rates vary by orders of magnitude. This variation blurs the relationship between crater size and lifetime, so that no serious information about the temporal variation in erosion rates can be gathered. The obtained erosion rates remain temporal mean values, and we can only try to specify the time interval of averaging or, more precisely, the sensitivity of the mean value as a function of the time before present.

The sensitivity of our estimated erosion rate with regard to time can be assessed using the ages of craters. As available information about the age of individual craters is often vague or only provides an upper or a lower limit, we compute the lifetime of each crater from the ratio of its depth (inferred from the diameter) and erosion rate. In order to avoid a bias by the local topography of the craters, we used the mean erosion rate of the respective province instead of the estimate at the location of the crater itself. We then use half of the estimated lifetime as an estimate of the age. Figure 7 gives the cumulative distribution of these ages. This distribution can also be interpreted as a sensitivity with regard to the time before present as it states how many of the existing craters would be affected by a change in erosion rate at a given time. It is immediately recognized that these sensitivity functions roughly decrease exponentially with time for all considered climatic zones as well as worldwide.

In order to obtain a robust estimate of the decay constant $\tau$, we use the time where the area below the curve from 0 to $\tau$ amounts to a fraction $1 - \exp(-1) \approx 63\%$ of the total area. This results in a minimum value of $\tau = 13$ Ma for the temperate zone and a maximum value of $\tau = 70$ Ma for the cold climate zone. So it is not possible to define a distinct time window of sensitivity for our method, but we find that the sensitivity exponentially decreases with time before present. As the worldwide mean erosion is dominated by the temperate zone and the tropical zone showing the smallest decay constant, we suggest $\tau = 13$ Ma as a conservative estimate. So our approach covers a time span characterized by a cooling climate, but without any fundamental changes in the location of the continents on Earth and in the spatial distribution of the orogens.

## 7    Has erosion globally increased?

Taking into account an exponentially decreasing sensitivity with $\tau = 13$ Ma, Fig. 8 compares our result on the worldwide long-term mean erosion rate with previous results. The green area represents our result of $r = 78$ m/Ma with the 70% con-fidence intervals. The decreasing opacity visualizes the exponentially decreasing sensitivity with $\tau = 13$ Ma.

Except for the average Pleistocene (2.58–0.01 Ma b.p.) erosion rate, our result is significantly higher than the estimates derived from preserved sediments for all epochs. All these estimates are even much below our lower 95% confidence limit of 52 m/Ma. This result supports the hypothesis of Willenbring and von Blanckenburg (2010) that the erosion rates obtained from preserved sediments are much too low.

As a reference value for the worldwide present-day erosion rate we use the values compiled by Willenbring et al. (2013). The studies starting from 1950 show a high variability from 35 m/Ma to 218 m/Ma. The mean value of these 31 studies is 76 m/Ma, and the standard deviation is 37 m/Ma, i.e., about 50% of the mean value. The standard deviation reduces if we consider only those 16 studies not older than 1975. We then obtain a mean value of 63 m/Ma with a standard deviation of 15 m/Ma. As these values do not change much if we reduce the data set further, we take $r = 63 \pm 15$ m/Ma as a reference value for the present-day erosion rate. As it is recognized in Fig. 8, the uncertainties in our long-term estimate and in the present-day erosion rate are similar, and the recent erosion rate is slightly below the lower bound of our 70% confidence interval. This means that we could reject the hypothesis of equal erosion rates at about 15% error level, but clearly not at 5% error level following the widely used practice in statistics. So our long-term estimate is even higher than the present-day erosion rates, but the uncertainty in the data does not allow the conclusion that the long-term erosion rates were indeed higher than the present-day rates, although this would be consistent with the retention of sediments in artificial reservoirs and with the tendency towards decreasing erosion in a cooling climate due to lower rates of weathering.

## 8    Potential systematic errors

The statistical variation arising from the sparse impact crater inventory on Earth already included in Fig. 3 is the most obvious source of uncertainty in our approach. However, there are also several potential sources of systematic errors that will be discussed in the following.

### 8.1    Impact craters as passive erosion markers

Our approach hinges on the idea that impact craters can be used as passive markers of large-scale erosion, although they may have a strong influence on local landform evolution in particular in an environment dominated by fluvial erosion (Wulf et al., 2019). In the first phase, the elevated crater rim will be eroded more rapidly than the surrounding region, and the crater could be filled by a lake depending on the climate. But when erosion in the surrounding region proceeds, the outflow of the river will incise into the rim, and the lake sediments will be eroded. When erosion finally reaches the deep-

est rocks altered by shock effects, the erosion of these last witnesses of the impact process should be tied by the rivers in the domain. Thus, the point where the structure cannot be proven as an impact crater any more should indeed be defined by the large-scale erosion of the region rather than the local processes in and around the crater.

However, the crater may indeed be invisible in the meantime due to local landform evolution, so that it may either not be listed as a crater exposed to the surface or might remain completely undetected. This loss of craters in the record may be relevant in particular for small craters and is addressed in the next section.

## 8.2 The completeness of the crater inventory

Estimated erosion rates and erosional efficacies are inversely proportional to the expected number of craters in our approach. Therefore, any incompleteness in the crater inventory directly leads to an overestimation of the erosion rate. Our paper on the crater inventory (Hergarten and Kenkmann, 2015) concluded that there is no evidence for a systematic incompleteness in the inventory above 6 km diameter. Comparing the real crater inventory with the prediction of a simple model based on erosion and age of the crust, it was shown that any significant incompleteness must cover the entire range of crater sizes above 6 km diameter. As small craters should remain undiscovered more easily than large craters, this was considered to be unlikely. Although the lack of newly detected craters listed in the Earth Impact Database supports this result further, there are probably still some undiscovered craters in the relevant range leading to some overestimation of the erosion rates.

In contrast to the proof of the impact origin by shock effects, the initial discovery of potential impact structures often relies on topography. So the question arises whether deeply eroded craters can really be detected in the topography until erosion reaches the deepest rocks altered by shock effects. The erosion of craters under fluvial conditions was recently simulated by Wulf et al. (2019), albeit over shorter time spans and with a different focus. Continuing these simulations over longer times, we found that a more or less circular drainage divide outside the crater is still visible when erosion reaches the crater floor. Nevertheless, the drainage network forming there may make it difficult to distinguish the crater from other structures as long as there is no distinct topographic signature and no variation in erodibility at the crater floor. However, all terrestrial craters larger than about 4 km in diameter are so-called complex craters with a central peak or an even more complex morphology of the crater floor. Thus, their characteristic topographic signature should not be erased so easily by eroding the crater floor, so that they should remain detectable even when erosion proceeds down the crater floor. We therefore expect the systematic error arising from an incompleteness in the range above 6 km in diameter to be much smaller than the statistical uncertainty.

The diameter range between 0.25 km and 6 km is more critical. Here the real record rapidly drops below the prediction at decreasing diameters. The discrepancy may be either due to an incompleteness in the record, but in principle it could also be possible that the protection of Earth from small impacts by the atmosphere is still underestimated in the model of Bland and Artemieva (2006). The value $I = 4.94 \times 10^{-5} \frac{\text{m}}{\text{Ma km}^2}$ used in this study already includes an empirical correction for this apparent incompleteness in the diameter range between 0.25 km and 6 km, so that it does not lead to a systematic error in itself. However, if it arises from an incomplete record, the incompleteness must be random and should not systematically differ among the climatic zones. Comparing the numbers of small craters to the number of large craters, we did not find any systematic variation. If the incompleteness is related to the potential invisibility discussed in Sect. 8.1, the lifetime of the craters must still be inversely proportional to the regional erosion rate. This seems to be reasonable, but cannot be proven as long as there is no model for this process.

As these considerations cannot exclude any bias arising from taking into account craters smaller than 6 km, we performed the same analysis for the craters larger than 6 km (with $I = 2.99 \times 10^{-5} \frac{\text{m}}{\text{Ma km}^2}$), but did not encounter any significant effect on the results except for a larger formal statistical uncertainty due to the smaller number of craters.

## 8.3 The value of the parameter $I$

Similarly to the potential incompleteness of the crater inventory, the parameter $I$ occurring in all our calculations has an immediate effect on all estimated erosion rates. According to Eq. (9) in Hergarten and Kenkmann (2015), it relies on the rate of crater production as a function of the diameter (Bland and Artemieva, 2006) and on a relationship between the average depth down to which the impact origin of a crater can be proven by altered rocks as a function of the diameter. The crater production rate should be well constrained except for the potential uncertainty at small diameters discussed in Sect. 8.2. The relationship for the depth used by Hergarten and Kenkmann (2015) was based on a limited set of data, so that it is probably more uncertain. However, the uncertainty arising from this relationship should be clearly smaller than the statistical uncertainty.

## 8.4 The role of the relief

As the lifetime of a crater is inversely proportional to the erosion rate, the majority of craters is found in regions with rather low erosion rates and thus with low to moderate relief. In turn, erosion is in sum dominated by a rather small part of the surface with high relief as illustrated in Fig. 6. Therefore, the relationship between erosion rate and relief (Eq. 4) plays a central part in transferring information from the crater inventory to high-relief regions where the record is

sparse. Although the linear relationship defined in Eq. (4) is consistent with early work of Ahnert (1970) and with the data presented in Fig. 1, this relationship may be a major source of systematic errors.

In order to assess the influence of the linearity of the relationship, we assume a more general power-law relationship of the form

$$r = s\Delta^b \tag{7}$$

and repeat the analysis for scaling exponents $b$ in the range $0 \leq b \leq 2$. The result is given in Fig. 9.

The erosion rates in general increase with increasing scaling exponent $b$. This is the expected behavior as the mean erosion rate is somewhat tied to regions with low relief due to the higher data density, while the estimate at high relief relies more on the relationship between relief and erosion rate. The potential bias is highly asymmetric; the worldwide mean erosion rates would be more than three times as high as our estimate for $b = 2$, while the minimum erosion rate occurring at $b = 0.31$ (57 m/Ma) would be only 27 % lower than our estimate. However, such a strong deviation from the linear relationship is not very realistic. To our knowledge all studies in this context either found linear or slightly convex ($b$ slightly above 1) relations (see, e.g., Summerfield and Hulton, 1994). This means that our approach perhaps underestimates the worldwide erosion rate slightly.

However, the relief also bears a potential of an overestimation going beyond the validity of the linear relationship for two reasons.

1. For deriving the worldwide mean erosion rates (Fig. 3c) from those of the predominantly erosive provinces (Fig. 3b), we assumed that the relationship between relief and erosion rate also holds for the other provinces. This procedure is based on the idea that these regions consist of erosive parts with the same value $s$ as the purely erosive provinces and parts dominated by sediment deposition. Assuming that the value of $s$ is valid for the entire region requires that the depositional parts have zero relief. However, even completely depositional areas have a (rather small) nonzero relief in reality, and this relief also contributes to the mean relief. Thus, the contribution of the not predominantly erosive provinces to worldwide erosion will be slightly overestimated.

2. Even more important, relief has changed through time at the million-year scale. If this change was spatially uniform, it would only affect the values of the erosional efficacies $s$, while the erosion rates would still be valid. Effects of glaciation should also not be very strong as the relief is measured at quite large scales (10 km). However, the formation or the decay of entire orogens would disturb the assumed relationship between present-day relief and long-term erosion. Then the relationship between these two properties would be

weaker than we assumed, and the effect would be similar to a concave relationship ($b < 1$). So the real erosion rate could then be lower than our value obtained from the linear relationship. In the worst-case scenario, there would be no correlation between the present-day relief and the long-term erosion rate everywhere, and the real erosion rate could drop to the value of 57 m/Ma mentioned above. However, this is unrealistic, and we expect the potential bias to be much smaller.

## 8.5  The subdivision into climatic zones

The subdivision of Earth's surface into the primary Köppen-Geiger classes of the present-day climate is probably the most obvious source of potential systematic errors. First, the erosional efficacy of each climatic class is still a harmonic mean value, any spatial variation within a climatic class will result in an underestimation of the erosional efficacy. Beyond this, the climate has changed during the considered time intervals, and even significant parts of Earth's surface have moved, so that the question for the consequences of a potentially inappropriate subdivision of Earth's surface arises.

In this section we will use a simple model for illustrating that an inappropriate subdivision of the surface will result in a systematic underestimation of the worldwide mean erosion rate, but never in a systematic overestimation. In the worst case, the improvement coming from the subdivision will be entirely lost, and we would end up at the harmonic mean value.

We start from the example used for illustrating the underestimation by the harmonic mean in Sect. 2. We assume that Earth's surface consisted of two domains of equal sizes with a high erosion rate $r_h = 120$ m/Ma in one domain and a low erosion rate $r_l = 30$ m/Ma in the other domain. Let us now assume that we subdivide the surface in two also equally sized domains, but we were not able to delineate them correctly, so that both regions contain a mixture of $r_h$ and $r_l$. Let $\lambda$ be the contribution of the wrong erosion rate, so that domain 1 consists of $(1 - \lambda)$ of $r_h$ and $\lambda$ of $r_l$ and vice versa for domain 2. Then the estimated erosion rates of both domains are given by the harmonic mean values

$$r_1 = \frac{1}{\frac{1-\lambda}{r_h} + \frac{\lambda}{r_l}} \quad \text{and} \quad r_2 = \frac{1}{\frac{\lambda}{r_h} + \frac{1-\lambda}{r_l}}. \tag{8}$$

The estimated worldwide mean erosion rate is the arithmetic mean of $r_1$ and $r_2$.

The results of this simple model shown in Fig. 10 reveal that any imperfection in the subdivision causes an underestimation of the mean erosion rate. In the worst case, the mean erosion rate drops to the harmonic mean erosion rate of 48 m/Ma, so that the improvement achieved by the subdivision is entirely lost. As expected, the difference between the erosion rates of the two regimes is partly shadowed if the subdivision is not perfect. As the harmonic mean is dominated by the lower value, the high erosion rate $r_h$ is strongly

underestimated from domain 1 by the imperfect subdivision, while the lower rate $r_l$ is only slightly overestimated from domain 2. In this example, even 10 % wrong contribution in each domain cost almost half of the improvement coming from the subdivision in total.

However, this example is somehow extreme as two strongly different regimes are mixed here, while the variations on Earth should be more gradual. Nevertheless, the subdivision of the surface into a limited number of domains will always retain a part of the underestimation coming from the harmonic mean being an inherent property of the approach. This underestimation will mainly effect the climatic zones with a high erosional efficacy.

### 8.6   Scale dependence of erosion rates

The discussion about potential systematic errors in erosion and sedimentation rates has been initiated by the fundamental paper of Sadler (1981) addressing the dependence of sedimentary records on the considered time scale. In this context we must distinguish whether a dependence of the rate on the covered time interval really exists or arises from the measurement. The latter would refer to a situation where the spatial distribution of the measurements is biased towards high erosion rates in the recent past. This effect has been, e.g., considered in the model of Ganti et al. (2016) where it was assumed that erosion takes place in distinct events, and measurements are only performed if there was an erosional pulse within a short time interval before present. Our approach is obviously invulnerable by this type of bias.

The situation considered by Schumer and Jerolmack (2009) is, however, more complex. In this study it was shown that a heavy-tailed distribution of hiatus lengths leads to a systematic decrease of erosion rates with time scale, while a heavy-tailed distribution of the sizes of erosional pulses leads to a systematic increase. However, it can be expected that the effect decreases when averaging over large spatial domains, so that our method should be less susceptible to such a bias than approaches based on individual points.

As long as there is no generally accepted model for the time-scale dependence of erosion rates often found, we cannot refute any susceptibility of our approach for such a bias completely, but there is at least no reason why it should be larger than in other methods.

### 8.7   Intermittent periods of sedimentation

Going a step beyond the occurrence of hiatuses in the erosional history discussed in the previous section, intermittent phases of sedimentation should also be taken into account as a potential source of errors. As our approach addresses time scales of several million years, we cannot assume that all provinces considered as predominantly erosive (shields, orogens, igneous provinces) have been purely erosive over the entire time span.

Let us for the moment consider craters of a given depth $H$ only, and let us assume that we are actually in a phase of erosion. Figure 11 illustrates the three types of behavior that could arise from intermittent periods of sedimentation. For the green curve, we would find those craters produced in the continuous time interval since the depth of burial of the actual surface has dropped below $H$ (horizontal dashed line). The period of recording is extended compared to a purely erosive situation, so that our estimated erosion rate will be lower than the average over the erosive phases. So it should be emphasized that our mean erosion rates are net rates where periods of deposition contribute negatively to the average, but this should not be considered as a bias.

Potential systematic errors are illustrated by the blue and red curves. The blue curve describes a scenario where sediment deposition took place long ago – a situation that has occurred quite frequently in the history of Earth. Then the period of recording is extended. As long as the old craters are also detected, the erosion rate will be underestimated. In turn, the red curve describes a situation where a depth of burial corresponding to the considered crater depth $H$ has never been reached. Then the period of recording is shortened, so that the mean erosion rate will be overestimated. However, since the depositional phase also contributes to the crater inventory, the erosional period must be quite short in order to generate a significant overestimation.

Recapitulating the sources of systematic errors considered in the previous sections, there are two sources with unique direction. The residual incompleteness of the inventory above 6 km in diameter (Sect. 8.2) leads to an overestimation, while the imperfection in the subdivision into climatic zones (Sect. 8.5) results in an underestimation. The assumed linear relationship between relief and erosion rate (Sect. 8.4) and intermittent periods of sedimentation (Sect. 8.7) may introduce a bias in both directions, but at least for the latter, underestimation appears to be more likely than overestimation. The other potential systematic errors should be small. So there should in sum be some tendency towards underestimation rather than towards overestimation.

### 9   Conclusions

Our study yields long-term mean erosion rates as a function of topography expressed in terms of the 10 km relief and climate represented by the primary Köppen-Geiger classes. While the huge variation of topography on Earth makes the biggest contribution to the worldwide variability of erosion rates, our results reveal a significant systematic dependence on climate in contrast to the results of several previous studies. We found a fivefold increase in erosional efficacy from the cold regimes to the tropical zone. Furthermore we found the temperate and arid climates to be very similar concerning their erosional efficacy. In this context it should be emphasized that our study relates long-term erosion rates on the

time scale of some tens of million years to present-day topography and climate. Our approach yields long-term erosion rates of these parts of the crust actually belonging to a certain climatic zone without taking into account the climatic history. So the values obtained for the erosional efficacies of the 5 considered climatic zones must be interpreted with some caution. Strictly speaking, they are lumped values averaging all influences beyond the topography for 5 parts of the land surface. As a result of this averaging, the difference in recent erosional efficacy of a given climate – if defined in a similar way – can be expected to be even higher than predicted by our method. Furthermore, the erosional efficacy of the arid climate being similar to the temperate climate does not refute the importance of water for erosion, but may be related to less dry conditions in the present arid zone in the geological history.

With regard to the worldwide erosion rates we obtained a mean value on the time scale of some tens of million years of 78 m/Ma which is much higher than previous estimates derived from preserved sediments. As discussed in Sect. 8, this estimate should even be rather too low than too high, although the systematical errors should in sum be smaller than the statistical uncertainty. This result supports the hypothesis of Willenbring and von Blanckenburg (2010) that the apparent increase in worldwide erosion may be an artefact of the sedimentary record and that the observed increase in some mountainous regions (Herman et al., 2013) probably related to the Pleistocene glaciation could be a regional effect with a limited worldwide relevance. It should, however, be kept in mind that our result on changes in erosion rates have been derived by comparing absolute values. As stated in Sect. 6, our method in itself is not able to detect changes in erosion rates directly, so that the existence of intermittent phases with high or low erosion rates cannot be refuted.

Our estimate of the long-term mean erosion rate is even about 25 % higher than the mean value of the worldwide present-day erosion rates published since 1975. This result is qualitatively consistent with a decrease of erosion with decreasing temperature due to lower rates of weathering and could also be related to the retention of sediments in artificial reservoirs. However, both our long-term erosion rates and the present-day rates have uncertainties in the order of magnitude of the difference. Therefore we can conclude that the erosion rates have clearly been higher than they seem from preserved sediments and that there is no evidence for any change in worldwide erosion rates on the scale of some tens of million years.

**Data availability.** Data for reproducing the results and generating additional figures are available at http://hergarten.at/supplement.zip (preliminary location during the review process).

# Appendix A: Confidence intervals for the estimated erosion rates

Equation (6) used for determining the erosional efficacies $s$ of the climatic zones only involves the total number of craters $n$ in the considered zone as a random variable. As this variable follows a Poissonian distribution, confidence limits are readily obtained from the respective cumulative distribution. This also holds for the mean absolute rates within each climatic zone according to Eq. (4). Only the worldwide mean erosion rate being the area-weighted mean of the individual rates,

$$r = \frac{\sum_i A_i r_i}{\sum_i A_i}, \tag{A1}$$

involves multiple random variables, so that confidence interval cannot be directly computed from a single statistical distribution. However, as shown in Fig. 3, the 70 % confidence intervals (corresponding to the standard deviation for a Gaussian distribution) are almost symmetric on a logarithmic scale. We therefore use half of the widths of these intervals as estimates of the individual errors $\delta \log_{10} r_i$ and compute $\delta \log_{10} r$ by Gaussian error propagation:

$$(\delta \log_{10} r)^2 = \sum_i \left( \frac{\partial \log_{10} r}{\partial \log_{10} r_i} \delta \log_{10} r_i \right)^2 \tag{A2}$$

$$= \sum_i \left( \frac{r_i}{r} \frac{\partial r}{\partial r_i} \delta \log_{10} r_i \right)^2 \tag{A3}$$

$$= \frac{\sum_i (A_i r_i \delta \log_{10} r_i)^2}{\left( \sum_i A_i r_i \right)^2}. \tag{A4}$$

Following the analogy of the 95 % confidence interval to twice the standard deviation for a Gaussian distribution, we define the 95 % confidence interval for the worldwide mean erosion rate by $2\delta \log_{10} r$. As the individual 95 % confidence intervals are more asymmetric and smaller than two times the 70 % confidence intervals on the logarithmic scale, this is a rather conservative estimate in the sense that the error towards lower erosion rate is overestimated.

**Author contributions.** S.H. designed the study and developed the theoretical framework and wrote the paper. T.K. provided the original idea and the knowledge on impact processes.

**Competing interests.** The authors declare that they have no competing interests.

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

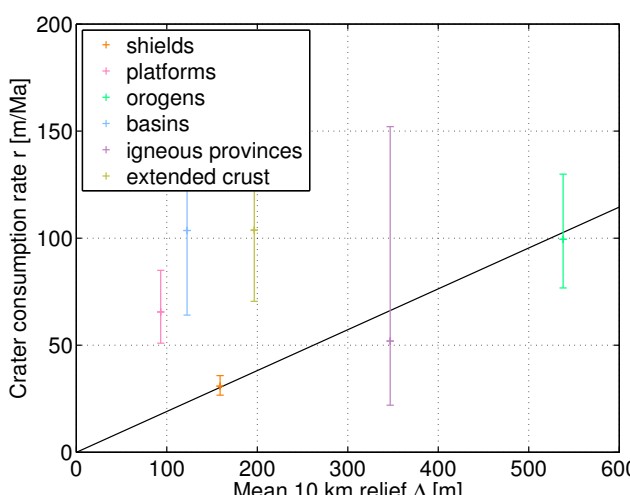

**Figure 1.** Rates of crater consumption derived from Eq. (2) vs. mean relief for the basic types of continental crust (http://earthquake.usgs.gov/data/crust/type.html). The error bars represent 70 % confidence intervals corresponding to the standard deviation for a Gaussian distribution.

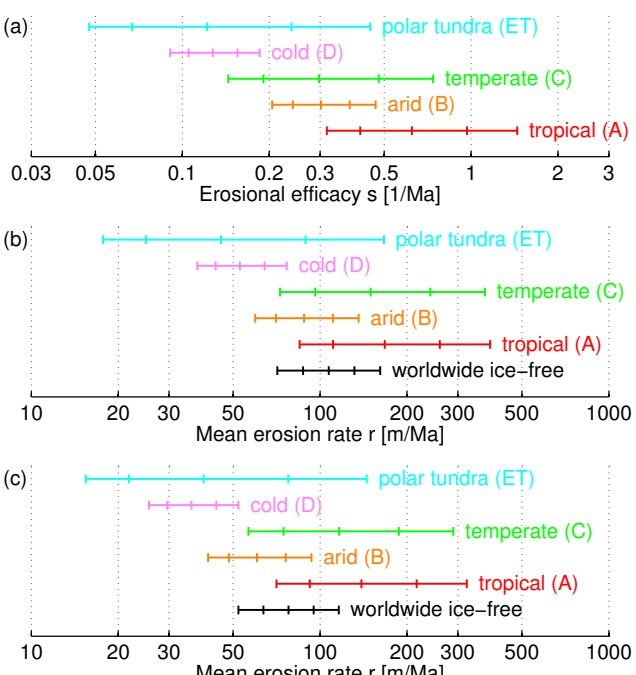

**Figure 3.** Erosion rates by climatic zones. (a) Mean erosional efficacies of the primary Köppen-Geiger classes. (b) Respective absolute mean erosion rates for the predominantly erosive provinces. (c) Absolute mean erosion rates extrapolated to the entire ice-free surface including the classes platform, basin, and extended crust. Error bars represent 70 % confidence intervals (corresponding to the standard deviation for a Gaussian distribution) and 95 % confidence intervals (see Appendix A).

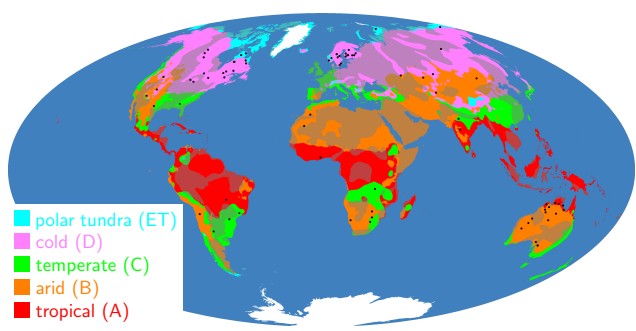

**Figure 2.** The primary Köppen-Geiger climate classes (Peel et al., 2007) considered in this study. Solid colors correspond to the predominantly erosive provinces (shield, orogen, igneous), while the respective pale colors mark those regions not considered in order to avoid a bias by sediment deposition. The black dots show the 77 craters with diameters $D \geq 0.25$ km located in the predominantly erosive provinces being the basis of our analysis.

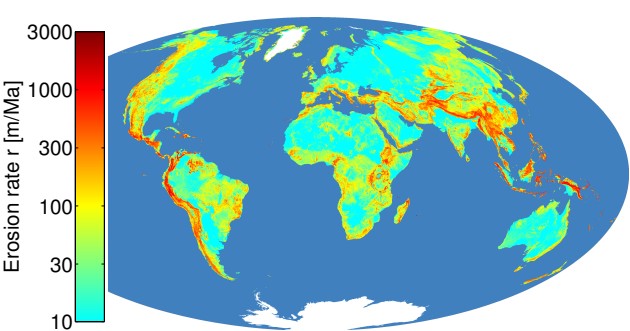

**Figure 4.** World map of the erosion rates obtained in this study.

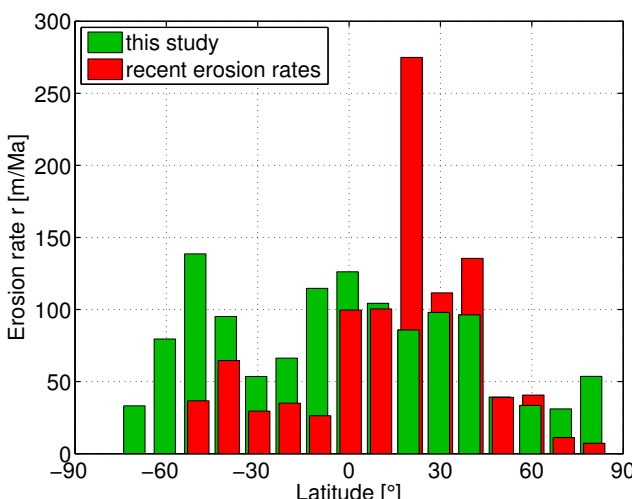

**Figure 5.** Mean erosion rates as a function of latitude in $10°$ intervals. Present-day erosion rates are taken from Wilkinson and McElroy (2007).

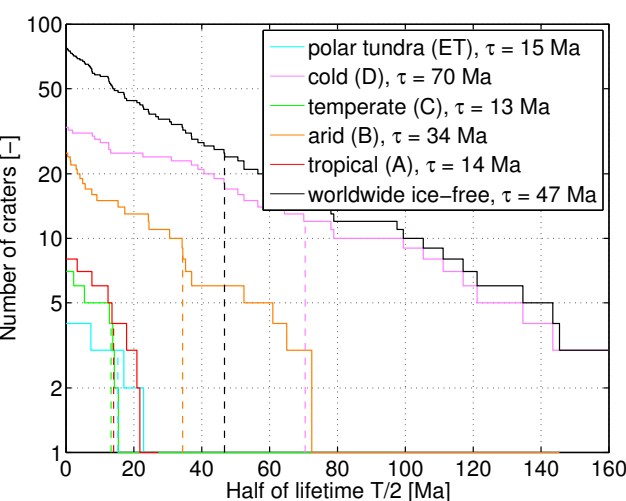

**Figure 7.** Cumulative number of the considered craters as a function of half of their estimated lifetime, equivalent to the sensitivity of the number of craters to changes in the erosion rate at a given time. The values of $\tau$ given in the legend are the decay constants of the exponential decrease.

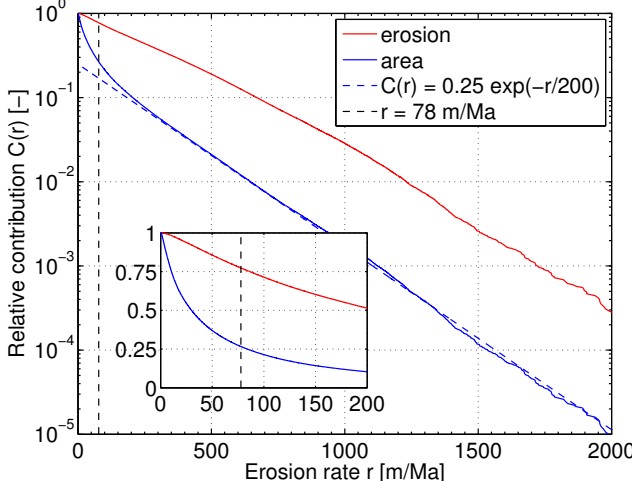

**Figure 6.** Cumulative distribution of the erosion rates and their contribution to total erosion. The blue curve shows the contribution of the part of the land surface with an erosion rate greater than $r$ to the total area, and the red curve its contribution to total erosion.

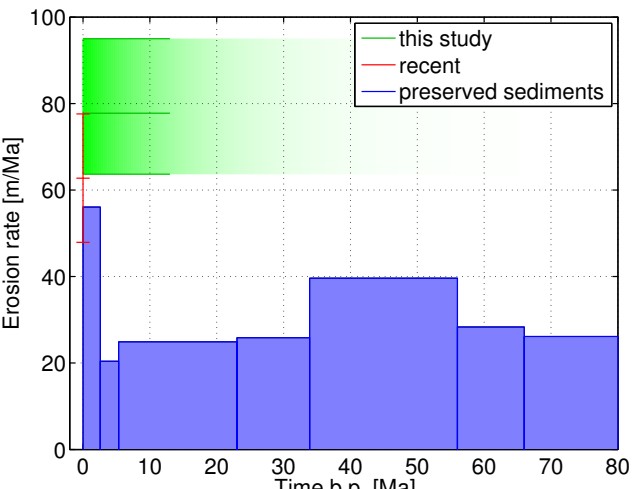

**Figure 8.** Comparison of our worldwide long-term mean erosion rate with estimates obtained from preserved sediments (Wilkinson and McElroy, 2007) and recent erosion rates compiled by Willenbring et al. (2013). The green area represents our result for the mean erosion rate of $r = 78$ m/Ma with $70\%$ confidence intervals. The decreasing opacity visualizes the exponentially decreasing sensitivity with $\tau = 13$ Ma.

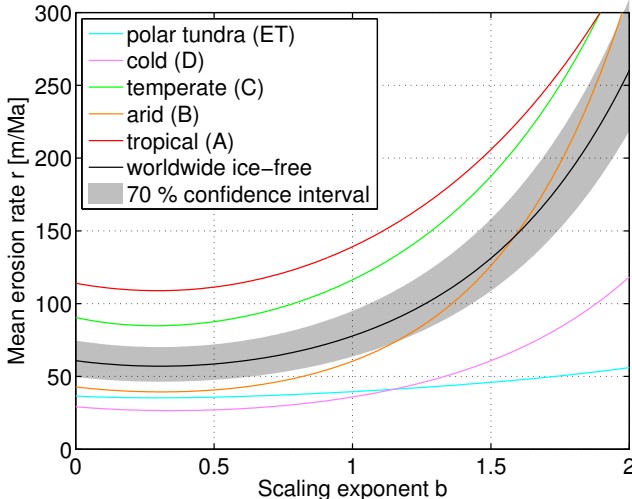

**Figure 9.** Results of our approach assuming a nonlinear relationship between relief and erosion rate (Eq. 7).

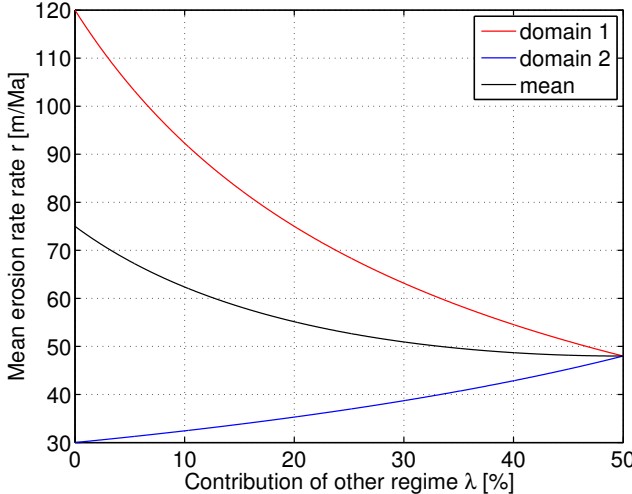

**Figure 10.** Estimated erosion rates if Earth consisted of two equally-sized parts with erosion rates $r_h = 120$ m/Ma and $r_l = 30$ m/Ma. It is assumed that two also equally-sized domains are analyzed separately, where the major part of domain 1 has the erosion rate $r_h$ and the major part of domain 2 has the erosion rate $r_l$, but each of the domains contains a given contribution $\lambda$ of the other erosion rate (Eq. 8).

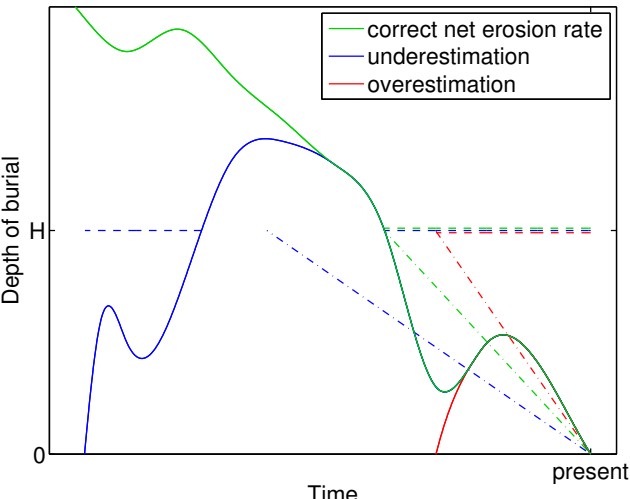

**Figure 11.** Three scenarios of intermittent phases of sedimentation. Solid curves: depth of burial of the present-day surface. Dashed lines: time intervals of crater production where a crater of a given depth $H$ would be detectable at the present surface. Dotted lines: equivalent erosional histories (same expected number of craters) with constant erosion rates.