# Peer review of "Has erosion globally increased? Long-term erosion rates as a function of climate derived from the impact crater inventory"

_Earth Surface Dynamics, 2018_

## Referee Comment (RC1) · Anonymous Referee #1 · 28 Sep 2018

I was invited to review this manuscript by the editors of ESurf. My expertise lays mostly in erosion processes themselves, though I have quite a bit of familiarity with the debate on apparent vs. real increases in erosion rates in the past, and I am also familiar with the key techniques used to determine erosion rates in the present and past (sedimentary fluxes, terrace preservation, cosmogenic nuclides -detrital or otherwise, thermochron, and sedimentary sequences) and their potential timescale biases. Finally I have only passing knowledge on crater formation and evolution as well as impact frequency. Though the results laid out here don't seem to depend critically on a detailed

understanding of these process, I am the first to admit a lack of knowledge in this area. Overall I feel well equipped to offer a well-rounded and critical review of this paper.

The authors propose to use the crater record on Earth as an independent measure of longterm erosion rates. This is a technique that has been applied on other bodies in the solar system to infer rates of surface activity. The major hurdles to applying this technique on Earth are the rapid rate of surface activity leaving only a fraction of the craters found on other bodies, and the influence that the atmosphere has on incoming objects. Given the previous work of these authors, it is clear that they are well positioned to overcome these challenges.

It is surprisingly challenging to determine erosion rates in the past, even with course resolution over large areas and time periods, and there is a major lack of consensus in the geomorphology community on this topic. A new technique, independent from existing ones is clearly welcome, and there is no question that this work is relevant. Overall, I find their approach to be original and exciting, and I support the effort whole-heartedly. My impression is that the work of the authors is thourough, careful and sophisticated when it concerns the frequency-magnitude distribution of impact craters and the completeness of the record. However, I feel that there are some shortcuts made to establish a connection between the crater record and the longterm erosion rate on the surface of the Earth.

Given the complexity, longevity and significance of the debate on past erosion rates, it is my opinion that there is an extra burden on the authors, and as it stands I do not recommend this paper for publication. However, I don't think the issues are insurmountable. The authors must do one of two things to convince me that this work should be published. They can either recast their results as very preliminary, and do more work to cast doubt on their own conclusions, describe all potential sources of error more thouroughly, and most importantly, discuss their results more conservatively. Or they can work to establish more convincingly some of the key assumptions that they make, and expand on how the processes that they depend on might actually function,

hopefully through a more complete literature search and some simply modelling.

I have 4 major issues with this work:

1. First, there should be more work to discuss how the crater record reflects erosion rates over long periods of time:

– In particular, is this approach really invulnerable to time-scale biases? Just stating that the record is spatially integrated doesn't convince me that there is no time-scale bias.

– What happens when erosion rates are spatially variable? This is dealt with later I know, but could be discussed more directly and clearly. The discussion of harmonic versus arithmetic mean is unclear and should be reworked for clarity.

– What happens when erosion rates are temporally variable?

– What happens if there are hiatuses that reflect a heavy tailed distribution as discussed in Ganti et al. 2016 - what if the hiatuses are spatially coherent? This probably isn't relevant for the global estimation, but what about when the authors divide the earth into more regions than there are craters in the record they use in the final analysis?

– What if erosion rates themselves follow a heavy tailed distribution, as discussed in Schumer et al., 2009?

2. There should be much more discussion about how craters actually erode away:

– Are the key processes the same for craters of all sizes? The largest craters modify the crust, leaving a mark in the rock over large areas, and it is clear that we will probably find them unless the crust is eroded to nearly the depth of the crater, or unless they are completely buried. Is this true of smaller craters? I would imagine that craters on the order of hundreds of meters to a few kilometers might be hidden more easily. Perhaps hillslope diffusion rates or soil production rates are the critical rates.

– Similarly, do small craters need to be completely eroded to disappear, or is it sufficient

**ESurfD**
to just erode them partly? This could lead to an overestimation of the longterm erosion rates. Either modeling or field results, potentially taken from the literature could be a major help here.

– Is there a regional bias that could effect the record of smaller craters? For example, could the North American ice sheets repeated advance and retreat have been sufficient to erase visible traces of craters below a certain size? Could something like this be responsible for the observed effect of climate on erosion rates through time? A better discussion of how craters of different sizes evolve and erode could guide the thinking here.

– Although I appreciate the urge to restrict the analysis to erosion only regions, over the timescales involved it seems to me that there may be no erosion only regions. There should be at the very least a larger concession to the error that sedimentation could introduce (see discussion for example in Willenbring et al., 2010).

3. The results of the climatic regions is interesting, but I am quite skeptical of this approach overall:

– Eastern Canada, Scandanavia and Australia seem to account for a majority of the craters used in this analysis (47 out of 77 or so). Can the authors bring in other lines of evidence to support the idea that these regions have been eroding more slowly than the rest of the Earth's surface for the last 10-100 Ma?

– Have the authors checked that there is no correlation between vegetation cover and crater frequency. Many of the places with many craters (northern canada, scandanavia and australia) are also regions that tend to have short or sparse vegetation.

– Though it is my impression that the authors have a good grasp of the appropriate statistics for this problem, I was plagued with questions about the role of chance while reading this paper. According to the authors, there are only 188 craters that have been found on earth, and of those only 112 are used in the analysis. Further, only 77

**ESurfD**
craters (as far as I can tell) fall in the erosion-dominated regions, though the authors then divide this into 89 sub-regions. My understanding then is that many of these subregions would have either 0,1 or at most 2 craters, and often the erosion rates will be optimized for the observation of finding no craters in the relevant region. How much error is introduced simply by the extraordinary rarity of having a significant event in a given region. According to Bland & Artemieva 2006, the expected time between craters > 500m is 20,000 years (I know the authors use 250m as the lower limit, but Bland and Artemieva give only the value for 500m craters). Assuming that impacts are truly randomly distributed on Earth, and that the surface area is 500,000,000 km^2, then it seems to me that the mean expected wait time between impacts >500m in a region of 1,000,000 km^2 would be on the order of 10 Ma. The expected time between craters > 500m for the smallest region they use would be greater than the age of the Earth (approx. 6 Ga). This temporal variability becomes significant when small regions are considered, and seems to me could lead to very large error bars on estimated erosion rates. Further the global erosion rates for the Polar Tundra, Temperate and Tropical regions are based on what appears to be only 4, 7 and 8 craters respectively. How does the estimated erosion rate change if there are one or two more (or fewer) craters in each climatic region?

– I think that a simple toy forward model could be extremely convincing here. It would be simple to build a model that randomly places craters down with the expected size and frequency on a large area with heterogeneous erosion rates that are known. Using the techniques applied here, the authors should show that the right answer can be recovered reasonably well when the crater record is a sparse as it is on Earth. They could further use the model to investigate the effect of temporally variable erosion rates on the inverted erosion rates.

– If the timescales of averaging are really approaching 100 Ma, what does it mean to divide the world into climatic zones? Over such timescales, not only did climate change significantly, but the crust itself was rearranged, moving craters from one climatic region

to another. The authors mention this, but these are described as effects that can blur the climate boundaries. I feel they don't acknowledge that plates can move 1000s of km and climate can change radically in such a timeframe.

– I think that the authors should consider removing this analysis overall, and focusing on the global rates, which are more convincing and also more relevant to the debate that they are addressing. However, a forward model would still be valuable!

4. My final issue concerns figure 9. I think that this figure is not an equal comparison of the two techniques. The marine sediment derived erosion rates are divided into different time periods while the crater-derived erosion rate is integrated over the history of the Earth. I think the authors miss what would be the single most significant test of the time-scale-bias-invulnerability of the crater-derived erosion rates that they claim. Because they have a record of craters with a wide range in sizes and because larger craters reach further back into time, it should be possible to subdivide their record in time instead of in space as they do for the climatic regions. Showing that the record reflects similar erosion rates for different size-groups of craters, and therefore over different time periods, would be a powerful piece of evidence in favour of their argument as well as a more accurate comparison of the crater record with the marine sedimentary record.

Details:

– Page 2, Lines 15-20: I think this is a bit of an unfair interpretation of previous work. High relief and high topography are both often the result of high uplift rates, and it is not surprising that they are correlated. Additionally, if relief is indeed the first order control on erosion rate, as you reasonably argue, then any comparisons of the influence of climate and lithology will have to take that into account. It would be necessary for example to show that the deviation from the expected linear trend is controlled by one of these two effects, or that for a given relief or slope the erosion rate is secondarily controlled by one of these factors. Studies such as Portenga and Bierman do not take
this into account. Some other studies that do find a clearer influence of climate (Ferrier et al. Nature 2013, Moon et al. Nature Geoscience 2011). I think it would be fair to use this reference to point out that climate is not a first order control on erosion rates, but not to imply that climate does not have the influence that we expect, as currently seems to be the implication.

– Page 2 line 30 to page 3 line 1: I think it would be important to express what I is and where it came from. I am guessing that $I = \int_{D_{ea}}^{D} \dot{N}'(D')H(D')dD' + H_{max}\dot{N}(D_{ea})$. I felt that I had to go back and read your previous paper before I understood equation 1, but it isn't referenced here. Even more critical would be an in depth discussion of the sources and magnitudes of error on I. What are the reasonable ranges of error. How much could it vary by? Perhaps with the least squares optimization it's a bit more complex, but my impression is that if I were 20% lower, the overall erosion rate would also be 20% lower. That seems like it would be a big deal.

– Page 3 line 2-3: This one line is a crux point in the paper, and I think is passed over a bit rapidly here. It is true that spatially averaged measurements will be less susceptible to the effects of temporal hiatuses and incomplete records that plague point measurements. However, there are other measures of erosion rate that are spatially integrated. The work of Herman et al, 2013 for example is based on thermochronological data which is integrated across tens of kilometers. More relevantly, Willenbring et al., 2010 mention 4 causes of the time-scale bias for sedimentary records some of which might matter in the case of craters, and they further show 4 data sets, several or all of which are spatially averaged, yet exhibit time-scale bias. More care should be given to demonstrate that the crater record is immune to time-scale biases.

– page 3, line 11-14: I don't think this point is made very well here. I guess you are trying to explain the difference between the old estimate of 59 m/Ma based on spatially homogenous erosion rates, and the new estimate of 78 m/Ma based on heterogenous rates? I think you should try to be a bit more clear on why exactly you are bringing in the harmonic and arithmetic means. Also, are you completely sure this is the correct

argument? What about in places where the erosion rate is based on the observation of no craters. Since you have no crater, you have no timescale, so it is not necessarily 'how long it takes to erode a given amount of material'.

– page 4, line 14 and other places: I think calling s the 'erosion rate per mean relief' is pretty awkward, I would jump straight to erosion efficiency as you eventually call it later in the manuscript

– Page 5, line 27: 'This result already suggests that erosion rates in the past might be much higher than those obtained from preserved sediments.' I feel that this point is way too strongly emphasized given the lack of discussion about potential sources of error in your estimate. I would remove it.

– Page 7, lines 15-19: I think this argument makes good sense for the timescale associated with the global erosion rates. However, for climate zone erosion rates, it seems to me that the timescales of the slower regions, e.g. the cold climate zone will be longer. This makes it harder to accept the idea that the climatic regions have any meaning over the integration timescales.

– Page 8, lines 5-7: Can you add some references for the widely accepted trend.

References:

– Willenbring, Jane K., and Friedhelm von Blanckenburg. "Long-term stability of global erosion rates and weathering during late-Cenozoic cooling." Nature 465.7295 (2010): 211.

– Herman, Frédéric, et al. "Worldwide acceleration of mountain erosion under a cooling climate." Nature 504.7480 (2013): 423.

– Ganti, Vamsi, et al. "Time scale bias in erosion rates of glaciated landscapes." Science advances 2.10 (2016): e1600204.

– Ferrier, Ken L., Kimberly L. Huppert, and J. Taylor Perron. "Climatic control of bedrock

river incision." Nature 496.7444 (2013): 206.

– Moon, Seulgi, et al. "Climatic control of denudation in the deglaciated landscape of the Washington Cascades." Nature Geoscience 4.7 (2011): 469 . – Schumer, Rina, and Douglas J. Jerolmack. "Real and apparent changes in sediment deposition rates through time." Journal of Geophysical Research: Earth Surface 114.F3 (2009).

---

## Referee Comment (RC2) · L. Goren (Referee) · 11 Oct 2018

The manuscript uses the world distribution of impact craters to estimate erosion rates. This is achieved by the assumption and the leading equation that describe the preservation potential of craters as a function of the erosion rate. The technique is used for evaluating erosion rates as a function of relief, erosion efficiency as a function of climatic zones, and global mean erosion rates. One of the main declared goals of the manuscript is to address the timely question of the effect of climate cooling on the mean erosion rate. Different studies came up with opposite conclusions concerning

the effect of cooling, and therefore, the motivation behind the study is well-justified.

—

Reading the abstract, I expected the analysis to be neat and simple, reading the rest of the text, I found it to be neat and very far from simple.

—

I identify five major methodological hurdles (the first two are probably the most important). Even if they can be dismissed, clarifications in the text are essential.

1. Could it be that craters are inherently more erodible than their surrounding due to the higher relief of the crater rim and the higher erodibility of the impact-induced breccia in and around the crater? If this is the case, then the time that it takes to erode a crater significantly underestimates the time that it takes to erode the surrounding material. This may introduce a strong bias toward the high erosion rates. The authors acknowledge (p. 6 lines 26—27) the effect of the local crater topography, but it is not further developed into an estimation of this potentially large bias.

2. Browsing through the supplementary material, it appears that in some cases, the statistics involve very small numbers, even in the erosive terrains. For example: 4 craters in cold orogens, 0 in cold igneous provinces, 4 in temperate shields, 2 in temperate orogens, 0 in tropical orogens, and so on. This raises the questions of: how do the authors estimate erosion rates in climatic-geologic terrains with 0 craters? Also, what is the validity of the estimation when the number of craters is so small? For the latter question, even a single unidentified/hidden crater (or a recently eroded crater) can have a significant impact on the statistics and the estimated erosion rates.

3. The authors discuss the possibility of terrains moving in between climatic zones during the relevant timescale. This discussion, however, is not sufficiently developed. For example the half-life is estimated for the different climate zones, but when a continent or a climate zone shifts, then this affects not only the erosion rate but also the half-life.

For example, if a continent has shifted from cold to temperate to tropic zones (I.e., India or Africa), then the half-life of the last climate zone should be even shorter.

4. On the same note, how can the effects of changing relief during the relevant timescale and the effect of quaternary glaciation be quantified?

5. The manuscript presents several biases for the estimation of the erosion rates, but their magnitudes are, in most cases, not evaluated. Even if currently it is not possible to evaluate the magnitudes, maybe the authors can explain what are the missing data and understanding that will allow their estimation in the future.

—

Some arguments, particularly those that are used for describing biases are quite hard to follow. For example:

1. Page 3. Lines 12-14. The point is clear, but readers might appreciate a simple artificial example.

2. Page 5. Lines 18-22. The text is too complicated.

3. Page 6. Lines 24-33 and 34-35. Hard to follow.

4. It is hard to interpret fig. 6. Consider adding an inset, where the y-axis is in percentage. (This might help the 75%-25% discussion).

—

Editing issues:

1. Sources for biases are presented throughout the manuscript in different sections. Organizing them in dedicated subsections might be helpful.

2. Missing commas after opening clauses.

3. Refer to appendices using the word 'appendix.'

4. Explain the vertical dashed black line in fig 6 in the captions.

—

Despite these substantial comments, and even if the methodology and the conclusions remain controversial, I believe that as long as all the uncertainties and biases are presented and discussed in the text (including the abstract and the summary), the manuscript could be an important addition to the global erosion rates discussion. I would have certainly liked to read it for its original methodology.

---

## Editor Comment (EC1) · J. Braun (Editor) · 2 Nov 2018

The two reviewers agree that the material contained in this manuscript is of great interest to the surface dynamics community because it proposes a new approach to investigate how erosion efficiency may have evolved through the recent geological past. Both reviewers, however, caution the authors that (1) they should be more explicit in giving robust estimates of the uncertainty attached to their estimate of how erosion rate has potentially changed over the past few tens of millions of years, and (2) they should better describe and justify a number of hypotheses made to reach their con-

clusion (preferential erodibility of craters, vegetation cover, motion of continents across latitudinal climatic zones, global change or redistribution of relief due to Quaternary glaciations, etc.) . There is also a need to better explain the approach and its statistical robustness (small number statistics, temporal vs spatial bias, effect of hiatuses, . . .) as, on the one hand, both reviewers appear to stumble on the same logical steps in the authors' arguments while, on the other hand, the authors argue that their approach is valid.

The authors acknowledge that both reviewers have performed a thorough and very constructive review of their manuscript. I fully concur with those statements. In view of the response given by the authors to the reviewers comments and arguments, I believe it is appropriate (and desirable) that the authors prepare a revised version of their manuscript. It should aim principally at improving the description of the method, the assumptions on which it is built and the uncertainty on the estimate of erosion rate evolution it provides. I also urge the authors to take into account all reviewers comments and suggestions to improve their manuscript for clarity and completeness, and, in doing so, enhance its potential impact.

---

## Author Response (AR1)

**Notes**

- Referee comments are marked with red color.
- Responses are identical with those in my original responses and are displayed in black.
- References to changes in the manuscript are marked in blue color; line numbers refer to the version with marked changes (latexdiff).

**Reviewer 1**

Dear Reviewer,

thank you very much for your thorough and constructive comments and for obviously spending a lot of time with our manuscript. It is always worrying if a reviewer even with an obviously solid theoretical background missed some of the main points of the approach. So we have to accept that we should explain the theory in more detail in order not to lose the majority of the readers too soon.

The problem is that the approach differs much from all other approaches and thus requires a quite specific mathematical / statistical treatment. In particular, the terrestrial crater record is so sparse that we have to take the big gun in order not to be killed by the statistical variation in the numbers of craters. So we will give our best to explain the following fundamental points more clearly in a revised version, namely

- why a subdivision into distinct zones (here the climatic zones) is necessary in order to overcome the shortcomings arising from the harmonic mean
- why the details of the subdivision (here the relationship between the present-day climatic zones and the paleoclimate) are not very important, and why even a completely wrong, subdivision of Earth's surface does not make any damage except that we got stuck at the harmonic mean and thus still underestimated the global erosion rate,
- how the parametric approach with the relief serves as a backbone to avoid the problem with the very small numbers in each province,
- and that all potential sources of error in sum indicate that the global erosion rate is still rather underestimated than overestimated, providing further support for our result that long-term global rates have been higher than previously assumed.

Extended and hopefully improved explanations at several places in Sect. 2, 3 and 4; removed Appendix A1 as it is no longer necessary with the new explanations.

Following the suggestion of the second reviewer we will also introduce a distinct section addressing the potential sources of errors.

New Sect. 8.

In detail:

I have 4 major issues with this work:

1. First, there should be more work to discuss how the crater record reflects erosion rates over long periods of time:

• In particular, is this approach really invulnerable to time-scale biases? Just stating that the record is spatially integrated doesn't convince me that there is no time-scale bias.

Our reasoning about a potential sampling bias did not refer to time-scale biases, but only to the potential bias by an uneven spatial location of sampling points, e.g., due to correlations between the number of outcrops and the topography. However, I think that taking values that are integrated over large spatial scales indeed reduce the potential time-scale bias (see the following points).

Page 3, lines 19–22 and Sect. 8.6.

• What happens when erosion rates are spatially variable? This is dealt with later I know, but could be discussed more directly and clearly. The discussion of harmonic versus arithmetic mean is unclear and should be reworked for clarity.

Seems that understanding the bias by taking the harmonic mean instead of the arithmetic mean if the erosion rates are spatially variable is indeed more difficult than I thought. I think we can explain it using an example in combination with the discussion of the subdivision into climatic zones.

Page 3, lines 23–31.

- What happens when erosion rates are temporally variable?
  - Nothing if the distribution of the erosion rates has a finite mean value and if there are no intermittent phases of deposition.
  - Erosion rates are underestimated (but never overestimated!) if there are intermittent phases of deposition.

Sections 8.6 and 8.7.

• What happens if there are hiatuses that reflect a heavy tailed distribution as discussed in Ganti et al. 2016 – what if the hiatuses are spatially coherent? This probably isn't relevant for the global estimation, ...

In the model proposed by Ganti et al. 2016, the observed time-scale basis does not really arise from the heavy-tailed distribution of the lengths of the hiatuses at least qualitatively. I tested this model with exponential and uniform distributions instead of the truncated Pareto distribution (which is also not heavy-tailed) and also discussed the results with Vamsi Ganti. The effect itself occurs in principle for all distributions as soon as you assume that all measurements where the erosion rate is zero are excluded. If you assume that zero erosion rates are also measurable and include them in the measurement, the effect completely vanishes for all hiatus distributions. Obviously impact craters do not mind if there was no erosion during the last years before present, so that our approach is definitely robust against the type of time-scale bias addressed by Ganti et al. 2016.

Section 8.6.

... but what about when the authors divide the earth into more regions than there are craters in the record they use in the final analysis?

This specific situation has no meaning at all; the problem that you are probably referring to already occurs if any region has zero crater count. Then the estimated erosion rate is infinite and thus destroying the whole estimate. Even if there are just a few craters in any region the error increases extremely due to the Poissonian statistics. This is the reason why only a small number of domains can be considered as completely independent (here the climatic zones). Relief as the primary control is included by a parametric approach in the form that the erosion rate is proportional to the relief, which means that all provinces belonging to the same climatic zone have the same ratio of erosion rate to relief. As shown in Appendix A, each climatic zone (not each province) is characterized by Poissonian statistics with 4 to 33 usable craters.

Page 4, lines 7-11, page 5, lines 16–19 and page 5, lines 26–19.

• What if erosion rates themselves follow a heavy tailed distribution, as discussed in Schumer et al., 2009?

Schumer et al. (2009) consider both heavy-tailed distributions of the hiatus lengths (in contrast to Ganti et al. 2016) and of erosional peaks. But in my opinion they do not consider a bias (due to measurement) but a real dependence of the mean erosion rate on the considered scale. If the hiatus lengths follow a heavy-tailed distribution, the erosion rate will tend towards zero in the limit of infinite time interval. If the erosional peaks follow a heavy-tailed distribution, the erosion rate will tend towards zero in the limit of infinite time interval. If the erosional peaks follow a heavy-tailed distribution, the erosion rate will tend towards infinity in the limit of infinite time interval. In both cases, erosion rate is no longer a well-defined term.

Going back to the results of our EPSL paper, the existence of such a scale dependence could even be refuted if the erosion rate was not spatially variable. A nonlinear scaling relation between time interval length and erosion rate would destroy the linear relationship between crater depth and lifetime, and this would be visible in Fig. 1 of the EPSL paper. However, as soon as the erosion rate is spatially variable, the effect may be blurred, so that we cannot refute the existence of a dependence of the erosion rate on the time scale. However, the effect should be the same (if present at all) for all methods or be weaker for methods averaging over large spatial scales such as our approach. I would therefore guess that a nonlinear scaling with time scale does not exist at large spatial scales, and that our approach is well-suited to avoid any time-scale bias.

Section 8.6.

- 2. There should be much more discussion about how craters actually erode away:
  - Are the key processes the same for craters of all sizes? The largest craters modify the crust, leaving a mark in the rock over large areas, and it is clear that we will probably find them unless the crust is eroded to nearly the depth of the crater, or unless they are completely buried. Is this true of smaller craters? I would imagine that craters on the order of hundreds of meters to a few kilometers might be hidden more easily. Perhaps hillslope diffusion rates or soil production rates are the critical rates.

This aspect was indeed briefly discussed in our EPSL paper on the completeness of the crater inventory. Following our key assumption that a crater remains visible until the regional erosion depth reaches the depth of the deepest altered rocks we found a really good fit above 6 km diameter, but the real record rapidly drops below the prediction at smaller diameters. Potential reasons are:

- (a) The record below 6 km diameter could still be incomplete (what was unfortunately considered as the key point in several press releases).
- (b) The protection of Earth from small impacts by the atmosphere is still underestimated in the model of Bland and Artemieva.
- (c) Small craters erode (or become invisible) faster than predicted by the regional erosion rate (the point you mention).

In our EPSL paper we even found an approximation for this apparent incompleteness, however, without being able to explain it physically or to decide which of the three

potential reasons applies here. The value I we used for the craters above 0.25 km in diameter includes this correction. This means that the apparent incompleteness of small craters does not introduce a systematic overestimation of the erosion rates. We only need to assume that the lifetime of small craters is still inversely proportional to the regional erosion rate, which is admittedly not completely clear, but does not really have a serious influence on the result. In order to test how much the small craters affect the results we applied the same method to the craters larger than 6 km some time ago, and we found no significant effect on the results except for a larger formal statistical uncertainty due to the smaller number of craters. For this reason we decided in include the small craters in the analysis (with the empirical correction). Sections 8.1 and 8.2.

• Similarly, do small craters need to be completely eroded to disappear, or is it sufficient to just erode them partly? This could lead to an overestimation of the longterm erosion rates. Either modeling or field results, potentially taken from the literature could be a major help here.

This point should be covered by our discussion of your previous point. Section 8.1.

• Is there a regional bias that could effect the record of smaller craters? For example, could the North American ice sheets repeated advance and retreat have been sufficient to erase visible traces of craters below a certain size? Could something like this be responsible for the observed effect of climate on erosion rates through time? A better discussion of how craters of different sizes evolve and erode could guide the thinking here.

We expected such variations during our work on this topic as there was some hope to be able to predict where undiscovered small craters could be found. However, we did not find any large regions where the number of small craters is either exceptionally high or exceptionally low in relation to the number of large craters. So we would tentatively claim that there is no such effect.

Section 8.2.

• Although I appreciate the urge to restrict the analysis to erosion only regions, over the timescales involved it seems to me that there may be no erosion only regions. There should be at the very least a larger concession to the error that sedimentation could introduce (see discussion for example in Willenbring et al., 2010).

Yes, there are indeed two potential effects.

- (a) If a region assumed to be erosional is a region of deposition over long times, craters are lost, so that the erosion rate is overestimated.
- (b) Phases of intermittent sediment deposition increase the lifetime of craters and thus result in an underestimation of the erosion rate.

In sum of both I would expect the second effect to be stronger, so that the erosion rate will be rather underestimated. Section 8.7.

- 3. The results of the climatic regions is interesting, but I am quite skeptical of this approach overall:
  - Eastern Canada, Scandinavia and Australia seem to account for a majority of the craters used in this analysis (47 out of 77 or so). Can the authors bring in other lines

of evidence to support the idea that these regions have been eroding more slowly than the rest of the Earths surface for the last 10-100 Ma?

At least for Australia we considered this in out first study on this topic (Lunar and Planetary Science Conference, 2014). Kohn et al. (2012, doi:10.1046/j.1440-0952.2002.00942.x) obtained a very low mean erosion rate of about 10 m/Ma over the entire continent at the 300 Ma scale from thermochronometry. As Fig. 4 shows, our estimate is quite close to this value for large parts of Australia. Taking the average over the entire continent we obtain about 26 m/Ma due to some regions with high relief, but taking into account the spatial variation and the different time scales I think that our estimate for Australia and also for other regions with a not too low number of craters should be quite ok.

No change to the manuscript as I think that it makes no sense to pick individual regions where our estimate matches the data from other studies particularly well.

- Have the authors checked that there is no correlation between vegetation cover and crater frequency. Many of the places with many craters (northern canada, scandanavia and australia) are also regions that tend to have short or sparse vegetation. How should this be checked formally? There is definitely a correlation between climate and vegetation, and nobody will seriously question a correlation between climate and erosion and thus between climate and crater record. A potential bias could only be detected in the inventory of small craters in relation to large craters. However, as mentioned above we did not find any evidence for such a bias so far. Section 8.2.
- Though it is my impression that the authors have a good grasp of the appropriate statistics for this problem, I was plagued with questions about the role of chance while reading this paper. According to the authors, there are only 188 craters that have been found on earth, and of those only 112 are used in the analysis. Further, only 77 craters (as far as I can tell) fall in the erosion-dominated regions, though the authors then divide this into 89 sub-regions. My understanding then is that many of these subregions would have either 0,1 or at most 2 craters, and often the erosion rates will be optimized for the observation of finding no craters in the relevant region. How much error is introduced simply by the extraordinary rarity of having a significant event in a given region. ...

This is obviously the problem of not getting the key point of the method correctly. As mentioned above, relief being the primary control is included by a parametric approach in the form that the erosion rate is proportional to the relief, which means that all provinces belonging to the same climatic zone have the same ratio of erosion rate to relief. As a consequence, only 5 independent parameters are fitted from the crater record (the erosion rate per relief = erosional efficacy s of each climatic zone). As shown in Appendix A, each climatic zone (not each province) is characterized by Poissonian statistics with 4 to 33 usable craters.

Page 4, lines 7-11, page 5, lines 16–19 and page 5, lines 26–29.

... According to Bland & Artemieva 2006, the expected time between craters  $\vdots$  500m is 20,000 years (I know the authors use 250m as the lower limit, but Bland and Artemieva give only the value for 500m craters). Assuming that impacts are truly randomly distributed on Earth, and that the surface area is 500,000,000 km2, then it seems to me that the mean expected wait time between impacts  $\vdots$ 500m in a region of 1,000,000 km2 would be on the order of 10 Ma. The expected time between craters  $\vdots$  500m for the smallest region they use would be greater than the age of the Earth

(approx. 6 Ga). This temporal variability becomes significant when small regions are considered, and seems to me could lead to very large error bars on estimated erosion rates. ...

This looks reasonable to me, but what is the consequence? Using our estimated erosion rates we find that highest expected number of craters among all regions is 16.1 (with 13 craters in reality), while the lowest expected number of craters among all regions is 0.0005 (with 0 craters in reality). We can, of course, include these numbers in the supplementary data sheet in order to make the numbers more convincing. However, as the statistics rely on the numbers per climatic zone, the numbers have no immediate meaning.

No change to manuscript.

... Further the global erosion rates for the Polar Tundra, Temperate and Tropical regions are based on what appears to be only 4, 7 and 8 craters respectively. How does the estimated erosion rate change if there are one or two more (or fewer) craters in each climatic region?

Yes, the crater counts follow Poissonian statistics, and the errors (70% and 95% confidence intervals) arising from this are given as error bars in Fig. 3. Not a big surprise that these error bars are quite large for the three climatic zones mentioned above, and also not a big surprise that these Poissonian statistics are the main source of uncertainty in the entire analysis.

Page 4, lines 7-11, page 5, lines 16–19 and page 5, lines 26–29.

• I think that a simple toy forward model could be extremely convincing here. It would be simple to build a model that randomly places craters down with the expected size and frequency on a large area with heterogeneous erosion rates that are known. Using the techniques applied here, the authors should show that the right answer can be recovered reasonably well when the crater record is a sparse as it is on Earth. ...

Not really. If you refer to different climatic zones, they are independent of each other. If you refer to different provinces within a climatic zone, they are constrained by the parametric relationship between relief and erosion rate. This means we already know the ratios of the erosion rates from the relief and only estimate one parameter. As mentioned above, this estimate is controlled by Poissonian statistics, and I do not think that it is very convincing to simulate Poissonian statistics with a numerical model.

No change to manuscript.

... They could further use the model to investigate the effect of temporally variable erosion rates on the inverted erosion rates.

Yes, but it is already clear that the obtained mean erosion rates are an average with a sensitivity decreasing exponentially through time into the past (Fig. 8). So the result would be that a high recent erosion rate has a stronger effect on the estimated mean rate than a high erosion rate in the past. But this specific model would not yield much more information.

No change to manuscript.

• If the timescales of averaging are really approaching 100 Ma, what does it mean to divide the world into climatic zones? Over such timescales, not only did climate change significantly, but the crust itself was rearranged, moving craters from one climatic region to another. The authors mention this, but these are described as effects that can blur the climate boundaries. I feel they dont acknowledge that plates can move 1000s of km and climate can change radically in such a timeframe.

Admittedly, this point was discussed in our manuscript only very briefly (page 5, lines 1-10). Starting from the point that the method applied to a single domains always yields a harmonic mean instead of the arithmetic mean value we always obtain a systematic underestimation as soon as the rates are spatially variable. A subdivision into a reasonable number of subdomains (so that the number of craters per domain is not too low) is the most convenient way to ship around this problem. As relief being the primary control is already covered by a parametric relationship (erosion rate proportional to relief), climatic zones are a somewhat natural choice.

From theory: If the erosional efficacy (erosion rate per relief) was constant within each climatic zone and also constant through time we would arrive at the correct result with regard to both the relationship between the climatic zones and the worldwide average (except for the statistical variation). This is probably not true. The other extreme would be completely random subdomains without any systematic differences in erosional efficacy. Then we would arrive at the same erosional efficacy on average within each domain (the harmonic mean over the respective domain). In total we would simply get stuck at the underestimation by the harmonic mean; the "wrong" subdivision would bring not progress at all, but also make no damage. Your argument is referring to the situation where the climatic zone make some sense, but they are probably not the perfect subdivision. The consequences are that

- the estimated erosion rates refer to the spatial domains corresponding to the actual climate zones (so not, e.g., to what is today arid climate over Earth's history),
- compared to the real erosional efficacies of the considered types of climate, the variation between our zones is smaller, and
- there is still some underestimation of the worldwide mean erosion rate.

I think these aspects could be explained in detail with a simplified consideration of two subdomains in the appendix, which would probably make much more sense than the toy model mimicking Poissonian statistics suggested above.

Section 8.5.

• I think that the authors should consider removing this analysis overall, and focusing on the global rates, which are more convincing and also more relevant to the debate that they are addressing. However, a forward model would still be valuable! Clearly not! We need any kind of subdivision of Earth's surface in order to avoid (or at least reduce) the underestimation of the mean erosion rate due to the harmonic mean. So the option would only be hiding the results referring to the climatic zones, and this makes no sense in my opinion.

No change to manuscript.

4. My final issue concerns figure 9. I think that this figure is not an equal comparison of the two techniques. The marine sediment derived erosion rates are divided into different time periods while the crater-derived erosion rate is integrated over the history of the Earth. I think the authors miss what would be the single most significant test of the time-scale-bias-invulnerability of the crater-derived erosion rates that they claim. Because they have a record of craters with a wide range in sizes and because larger craters reach further back into time, it should be possible to subdivide their record in time instead of in space as they do for the climatic regions. Showing that the record reflects similar erosion rates for different size-groups of craters, and therefore over different time periods, would be a powerful piece of evidence in favour of their argument as well as a more accurate comparison of the crater record with the marine sedimentary record.

This is basically true, but unfortunately it is practically impossible. I tried this some time ago. If the erosion rate was spatially homogeneous, then a characteristic time scale could be assigned to each crater depth. Then the problem would still be that all craters are sensitive to a time interval from the present, so that the inversion of the crater-size distribution is already somewhat unstable. But as the erosion rate varies by orders of magnitude due to the variation in relief alone, there is no realistic chance to invert the crater-size distribution with regard to time. And I agree that Fig. 9 is not a perfect representation of the two different methods, but I have no better idea and think the bar with a fading background color indicating the decreasing sensitivity is not too bad.

Page 9, lines 6–17; Fig. 9 (now Fig. 8) unchanged.

**Details:**

• Page 2, Lines 15-20: I think this is a bit of an unfair interpretation of previous work. High relief and high topography are both often the result of high uplift rates, and it is not surprising that they are correlated. Additionally, if relief is indeed the first order control on erosion rate, as you reasonably argue, then any comparisons of the influence of climate and lithology will have to take that into account. It would be necessary for example to show that the deviation from the expected linear trend is controlled by one of these two effects, or that for a given relief or slope the erosion rate is secondarily controlled by one of these factors. Studies such as Portenga and Bierman do not take this into account. Some other studies that do find a clearer influence of climate (Ferrier et al. Nature 2013, Moon et al. Nature Geoscience 2011). I think it would be fair to use this reference to point out that climate is not a first order control on erosion rates, but not to imply that climate does not have the influence that we expect, as currently seems to be the implication.

It was not our intention to say that climate has no effect on erosion; the message should have been that other controls (mainly relief) may shadow the effect of climate at large scales, so that it is quite difficult to quantify the effect of climate. When writing the paper we originally decided not to include the two papers as they refer more to the regional scale than to the worldwide scale. We will include them an write the discussion about shadowing the climatic effect by topography a bit more precisely.

Page 2, lines 14–16 and 22–25.

• Page 2 line 30 to page 3 line 1: I think it would be important to express what I is and where it came from. I am guessing that  $I = \int_{D_{ea}}^{D} \dot{N}(D)H(D)dD + H_{max}\dot{N}(D_{ea})$ . I felt that I had to go back and read your previous paper before I understood equation 1, but it isnt referenced here. ...

Your guess is correct; we will explain this a bit more in detail and reference the equation.

Page 3, lines 9–11.

 $\dots$  Even more critical would be an in depth discussion of the sources and magnitudes of error on I. What are the reasonable ranges of error. How much could it vary by? Perhaps with the least squares optimization its a bit more complex, but my impression is that if I were 20% lower, the overall erosion rate would also be 20% lower. That seems like it would be a big deal.

Your guess with the effect of a 20 % variation is correct. However, the value of I originates from the crater production rate and the depth-diameter relation of craters (both being quite well constrained and described in our EPSL paper). I am quite sure than the uncertainty in I is much smaller than the statistical errors due to the small Poissonian crater counts already included in Fig. 3.

Section 8.3.

• Page 3 line 2-3: This one line is a crux point in the paper, and I think is passed over a bit rapidly here. It is true that spatially averaged measurements will be less susceptible to the effects of temporal hiatuses and incomplete records that plague point measurements. However, there are other measures of erosion rate that are spatially integrated. The work of Herman et al, 2013 for example is based on thermochronological data which is integrated across tens of kilometers. More relevantly, Willenbring et al., 2010 mention 4 causes of the time-scale bias for sedimentary records some of which might matter in the case of craters, and they further show 4 data sets, several or all of which are spatially averaged, yet exhibit time-scale bias. More care should be given to demonstrate that the crater record is immune to time-scale biases.

This is partly true, but in principle this aspect has been discussed above.

Section 8.6.

• page 3, line 11-14: I dont think this point is made very well here. I guess you are trying to explain the difference between the old estimate of 59 m/Ma based on spatially homogenous erosion rates, and the new estimate of 78 m/Ma based on heterogenous rates? I think you should try to be a bit more clear on why exactly you are bringing in the harmonic and arithmetic means. Also, are you completely sure this is the correct argument? ...

Definitely!

Page 3, lines 23–31.

... What about in places where the erosion rate is based on the observation of no craters. Since you have no crater, you have no timescale, so it is not necessarily how long it takes to erode a given amount of material.

Also discussed above.

Page 4, lines 7-11, page 5, lines 16–19 and page 5, lines 26–29.

• page 4, line 14 and other places: I think calling s the erosion rate per mean relief is pretty awkward, I would jump straight to erosion efficiency as you eventually call it later in the manuscript

Good idea! The only thing I am not completely sure is whether we should use efficiency instead of efficacy. We could indeed see relief as a resource and interpret s in the sense how efficiently the climatic zone generates erosion from using relief. However, I still feel that efficacy could be more appropriate than efficiency.

Adjusted throughout the manuscript starting from page 6, line 5.

• Page 5, line 27: This result already suggests that erosion rates in the past might be much higher than those obtained from preserved sediments. I feel that this point is way too strongly emphasized given the lack of discussion about potential sources of error in your estimate. I would remove it.

The phrase "already suggests that" was chosen taking into account that the timescale has only been roughly estimated at this point and that there was no thorough discussion of the potential errors. I think it should stay there as a preliminary conclusion at the end of the section.

No change to manuscript.

• Page 7, lines 15-19: I think this argument makes good sense for the timescale associated with the global erosion rates. However, for climate zone erosion rates, it seems to me that the timescales of the slower regions, e.g. the cold climate zone will be longer. ...

This is true, and the time scale for the slowest zone is even given in line 14 on the same page, while the time scales for the other zones are given in Fig. 8.

No change to manuscript.

... This makes it harder to accept the idea that the climatic regions have any meaning over the integration timescales.

Although there was a bit more shift over the longer time scale, this does not affect the meaning of the climatic zones as discussed above.

Section 8.5.

• Page 8, lines 5-7: Can you add some references for the widely accepted trend.

Some references were given in the introduction, but I do not mind rewriting this sentence. Page 10, lines 26–27.

**Reviewer 2**

Dear Liran Goren,

thank you very much for your thorough and constructive comments. I am quite sure that we will be able to submit an improved version of the manuscript soon.

...Reading the abstract, I expected the analysis to be neat and simple, reading the rest of the text, I found it to be neat and very far from simple.

Both reviews have indeed convinced me that there are several points that are not as simple as I thought. The problem is that the approach differs much from all other approaches and thus requires a quite specific mathematical / statistical treatment. In particular, the terrestrial crater record is so sparse that we have to take the big gun in order not to be killed by the statistical variation in the numbers of craters. So we will give our best to explain the methodological aspects more clearly in order not the lose the majority of the readers too soon.

Extended and hopefully improved explanations at several places in Sect. 2, 3 and 4; removed Appendix A1 as it is no longer necessary with the new explanations.

I identify five major methodological hurdles (the first two are probably the most important). Even if they can be dismissed, clarifications in the text are essential.

1. Could it be that craters are inherently more erodible than their surrounding due to the higher relief of the crater rim and the higher erodibility of the impact-induced breccia in and around the crater? If this is the case, then the time that it takes to erode a crater significantly underestimates the time that it takes to erode the surrounding material. This may introduce a strong bias toward the high erosion rates. The authors acknowledge (p. 6 lines 26-27) the effect of the local crater topography, but it is not further developed into an estimation of this potentially large bias.

I think this will indeed be the case for most of the craters, but it will not introduce a major bias. In the first phase, the elevated crater rim will perhaps be eroded more rapidly, and the crater could be filled by a lake. As erosion in the surrounding region proceeds, the outlet of the river will incise, and the lake sediments will be eroded. Finally the lower

bound of the altered rocks at the crater floor will be reached, and at this point the erosion of the crater floor should be tied by the rivers in the domain, so that the point where the crater cannot be detected or proven any more should indeed be defined by the large-scale erosion of the region.

Section 8.1.

2. Browsing through the supplementary material, it appears that in some cases, the statistics involve very small numbers, even in the erosive terrains. For example: 4 craters in cold orogens, 0 in cold igneous provinces, 4 in temperate shields, 2 in temperate orogens, 0 in tropical orogens, and so on. This raises the questions of: how do the authors estimate erosion rates in climatic-geologic terrains with 0 craters? Also, what is the validity of the estimation when the number of craters is so small? For the latter question, even a single unidentified/hidden crater (or a recently eroded crater) can have a significant impact on the statistics and the estimated erosion rates.

At this point both reviewers got stuck, so it is probably the point with the highest need for a better explanation. In the first step, relief was assumed (and roughly verified) as the primary control on erosion, and a linear relationship was established (for the predominantly erosive provinces). Then a subdivision into the climatic zones was performed in order to take into account climate as the secondary control, but keeping the linear relationship between relief and erosion rate in each zone. As a consequence, only 5 independent parameters are fitted from the crater record (the erosion rate per relief = erosional efficacy s of each climatic zone). These parameters follow Poissonian statistics per climatic zone (not per province). So statistics indeed relies on only 4 craters for the polar tundra class (reflected in very high error bars in Fig. 3), but this class does not contribute very much to worldwide erosion, while the numbers are higher in the other classes.

Page 4, lines 7-11, page 5, lines 16–19 and page 5, lines 26–29.

3. The authors discuss the possibility of terrains moving in between climatic zones during the relevant timescale. This discussion, however, is not sufficiently developed. For example the half-life is estimated for the different climate zones, but when a continent or a climate zone shifts, then this affects not only the erosion rate but also the half-life. For example, if a continent has shifted from cold to temperate to tropic zones (I.e., India or Africa), then the half-life of the last climate zone should be even shorter.

This is in principle true and applies to both the erosional efficacy (and thus the erosion rates) and the half-lives. Both estimates refer to the part of the crust that corresponds to the respective climatic zone today. For your example this means that our estimates for the tropical zone do not completely reflect tropical conditions, but are a mixture of tropical with some contribution of cold climate during history. And as you mention, the contribution from the cold zone is even an average over a longer time span than the main contribution (tropical zone). But in principle the only consequence is that the statistical distribution of the half-lives within each climatic zone shown in Fig. 8 (exponential distribution) may not be completely random, but may have a systematic spatial variation.

I would say the more important aspect in this context is the effect on the erosion rates themselves. Here we will add an explanation (probably a section in the appendix) what happens if the subdivision into climatic zones does not reflect the climatic conditions over the geological history properly. In this case, the estimates for the chosen zones are closer to each other than they would be if the choice of the zones was perfect, and the worldwide mean erosion rate will be underestimated (closer to the harmonic mean), but never be systematically overestimated. Section 8.5.

4. On the same note, how can the effects of changing relief during the relevant timescale and the effect of quaternary glaciation be quantified?

As far as I can see, this could be the only source of significant systematic errors (in relation to the statistical uncertainty that is already quite high as shown in Fig. 3) that could be realistically expected. If the ratios of the average relief have changed significantly over the history, the erosional efficacies will indeed be biased. However, the effect finally cancels when moving from the efficacies to erosion rates. Nevertheless, the erosional efficacy would indeed be overestimated if the relief in a climatic zone was significantly reduced recently, e.g., by glaciation. Trying to avoid such effects was indeed the main reason for taking the relief over quite large spatial windows, so that, e.g., the shape of individual valleys has no effect.

Section 8.4.

5. The manuscript presents several biases for the estimation of the erosion rates, but their magnitudes are, in most cases, not evaluated. Even if currently it is not possible to evaluate the magnitudes, maybe the authors can explain what are the missing data and understanding that will allow their estimation in the future.

Yes, it is indeed difficult to quantify the biases or uncertainties, but nevertheless we will discuss them in more detail in a revised version.

- (a) Uncertainties arising from the depth-diameter relation and from the crater production function are probably quite low and negligible in relation to the statistical uncertainty.
- (b) The completeness of the available crater record may be a more critical point. Any systematic incompleteness of the record linearly transforms to an overestimation in the erosion rates. In our EPSL paper we have only shown that, if there is a significant incompleteness, it must extend uniformly over the entire diameter range above 6 km and concluded that this is unlikely.
- (c) The linear relationship between relief and erosion rate might even be the most critical point. According to the relationship between lifetime and erosion rate, most of the information is drawn from regions with low to moderate erosion rates, while regions with high erosion rates also contribute much to worldwide erosion. If the erosion rate increases more than linearly with relief in reality, we will underestimate the worldwide erosion rate and vice versa. We could indeed add some estimate on the magnitude of this potential bias.

Section 8 and subsections.

Some arguments, particularly those that are used for describing biases are quite hard to follow. For example:

1. Page 3. Lines 12-14. The point is clear, but readers might appreciate a simple artificial example.

Indeed, we will combine this with the discussion of the subdivision into subdomains.

Page 3, lines 23–31.

2. Page 5. Lines 18-22. The text is too complicated.

 $\operatorname{Hm}\,\ldots\,\, ok$

Page 7, lines 19–25.

3. Page 6. Lines 24-33 and 34-35. Hard to follow.

 $\operatorname{Hm}\,\ldots\,\operatorname{ok}$

Removed (page 8, line 29 to page 9, line 9.) and replaced by a hopefully better discussion in Sect. 8.4.

4. It is hard to interpret fig. 6. Consider adding an inset, where the y-axis is in percentage. (This might help the 75%-25% discussion).

Good idea, we will do this unless we find an even better solution.

Done.

Editing issues:

1. Sources for biases are presented throughout the manuscript in different sections. Organizing them in dedicated subsections might be helpful.

I think it would indeed be a good idea to do this or even to make one dedicated section on this topic.

Section 8 and subsections.

2. Missing commas after opening clauses.

I checked the text and indeed found some.

- 3. Refer to appendices using the word appendix. Fixed (page 7, line 29); the other appendix has been removed.
- 4. Explain the vertical dashed black line in fig 6 in the captions. Added to the legend instead of to the caption.

Should be no problem to fix these points, thanks!

Despite these substantial comments, and even if the methodology and the conclusions remain controversial, I believe that as long as all the uncertainties and biases are presented and discussed in the text (including the abstract and the summary), the manuscript could be an important addition to the global erosion rates discussion. I would have certainly liked to read it for its original methodology.

Thanks! I hope that the readers of the final papers will also like to read it.

All the best, Stefan

[revised manuscript text omitted]
 (taking into account the incompleteness) in a given region of area A at an erosion rate r is

$$\underline{nN} = \frac{AI}{\underline{r}} \frac{I}{\underline{r}}$$
(1)

with a constant  $I = 4.94 \times 10^{-5} \frac{\text{m}}{\text{Ma} \text{km}^2}$ . Equation 2 can be used to estimate the long-term mean erosion ratefrom the number of impact craters . As this method directly yields some spatially (Hergarten and Kenkmann, 2015, Eq. 9). The value of *I* takes into account the crater production rate, the depth-diameter relation of craters and an estimate of the potential incompleteness of the inventory in the diameter range from 0.25 km to 6 km.

If erosion is spatially homogeneous in the considered domain, Eq. (1) immediately predicts the expected number n of craters

as

5

10

$$n = AN = \frac{AI}{r} \tag{2}$$

where A is the size of the domain. For heterogeneous erosion, Eq. (1) yields

15
$$n = \int N dA = I \int \frac{1}{r} dA.$$
 (3)

Inverting this relationship allows for an estimation of some spatially and temporally averaged erosion rate , it avoids any sampling bias occurring in other methods based on point-like measurements. from the number of impact craters in a given region.

At this point it is noteworthy that the spatial average is not an average over the locations of the existing craters, but over the entire area. In other words, the approach does not only derive information on erosion rates from regions where craters are, but also from crater-free regions. It therefore avoids the potential sampling bias due to an uneven distribution of locations that

20

might occur in all methods where erosion rates measured at points or over small areas must be transferred to large areas. In turn, an inevitable statistical uncertainty arises from the low number of the occurrence of r in the denominator in Eq. (3) reveals that the number of craters in a given region does not yield the arithmetic mean erosion rate (as it is relevant, e.g., for the

- 25 sediment yield), but the harmonic mean rate. The latter is always lower than the arithmetic mean, and the discrepancy increases with increasing spatial heterogeneity. Let us illustrate the difference by a simple example (which will be revisited in Sect. 8.5). If the entire surface of Earth consisted of two parts of equal sizes where one part has a high erosion rate of  $r_h = 120$  m/Ma and the other a low erosion rate  $r_l = 30$  m/Ma, the arithmetic mean rate would be 75 m/Ma. The harmonic mean erosion rate would, however, be only  $(\frac{1}{2}(r_h^{-1} + r_h^{-1}))^{-1} = 48$  m/Ma and thus be more than one third lower than the arithmetic mean rate.
- 30 Taking this discrepancy into account, it can be expected that the harmonic mean rate for the entire ice-free land surface of r = 59 m/Ma obtained by Hergarten and Kenkmann (2015) significantly underestimates the arithmetic worldwide mean.

Overcoming this limitation is one of the main goals of this paper. Subdividing the total surface into a sufficient number of domains and then averaging over these domains seems to be a straightforward idea, but is limited by the low number of impact craters exposed at Earth's surface. At the time of the original study, the Earth Impact Database (http://www.passc.net/EarthImpactDatabase/) comprised 188 terrestrial craters in total with only 112 of them exposed at the surface and wider than 0.25 km. While two more

- 5 craters have been added to the database until now, the number of relevant craters is still  $\frac{112}{112}$ , so that the value of I given above is still valid. Due to this low total number, the approach is most suitable for large regions. 112. Application to the entire ice-free surface of Earth yields a worldwide mean crossion rate of r = 59 m/Ma (Hergarten and Kenkmann, 2015). While this number in total provides a moderate statistical error of about 10% (standard deviation of Poisson distribution), the statistical errors rapidly increase if the number of craters per domain decreases. In particular, crater-free domains would cause serious
- problems as the estimated erosion rate would be infinite (with an infinite error, too). Therefore, a more sophisticated approach 10 is required; it will be explained in the following sections.

Although this method is not susceptible to sampling errors, spatial heterogeneity introduces a bias since the approach is based on lifetimes of craters instead of crosion rates, reflected in the occurrence of r in the denominator of Eq. 2. In other words, it is measured how long it takes to erode a given amount of material and not how much material is eroded in a given

- time span. As a consequence, applying Eq. 2 to a region with a non-uniform erosion rate yields the harmonic mean rate being 15 always lower than the arithmetic mean, resulting in an underestimation of the mean rate. In turn, craters 
[revised manuscript text omitted]

- 20 relief and erosion found for the predominantly erosive provinces holds there, too. This procedure is appropriate if the not predominantly erosive provinces consist of erosive parts and parts parts of these provinces are erosive with the erosional efficacy of the respective climate zone, while the rest is dominated by sediment deposition with. Assuming that regions of sedimentation have a very small (strictly speaking, zero) relief. Otherwise, the erosion rates given in Fig. 3e may be slightly biased towards high values., the erosional efficacy is also valid for these mixed zones and thus for the entire climate zone
- 25 (including the regions of sedimentation). Depending on the climate class, the mean erosion rates decrease by 13 % to 32 % due to the lower mean relief of the extrapolated provinces. However, the results are qualitatively similar to those obtained for the predominantly erosive provinces.

The area-weighted mean over the five climatic zones (Fig. 3c) yields a worldwide mean erosion rate of r = 78 m/Ma (107 m/Ma for the predominantly erosive provinces) with 95 % confidence limits of 52 m/Ma and 116 m/Ma (see ??Appendix A).

30 Our result is almost 40 % higher than the mean Pleistocene (2.58–0.01 Ma b.p.) erosion rates of r = 56 m/Ma obtained from preserved sediments (Wilkinson and McElroy, 2007). The latter value is even close to our lower 95 % confidence limit, and all known values for earlier periods of Earth's history are even lower. This result already suggests that erosion rates in the past might be much higher than those obtained from preserved sediments. We will return to this point after considering the time scale addressed by our approach more thoroughly (Sect. 6).

**5 The spatial distribution of erosion on Earth**

Figure 4 shows a world map of the estimated erosion rates using the 10 km relief on a  $0.1^{\circ} \times 0.1^{\circ}$  lattice and the values *s* of the respective climate zones. The dominance of topography over climate is immediately visible. While the mean relief amounts to 260 m, the maximum relief is 5887 m, which is more than 20 times larger than the mean relief. In contrast, the erosional

efficacy s differs only by about a factor of 5 between the warmest and the coldest climate classes. However, very high erosion

5

rates above 1000 m/Ma occur over considerable areas only in combination of tropical climate and high relief. The largest domain with estimated erosion rates above 1000 m/Ma is found in New Guinea.

Figure 5 compares the estimated erosion rates with the present-day erosion erosion rates published by (Wilkinson and McElroy, 2007) Wi on the study of Ludwig and Probst (1996). As this study focused on organic carbon, specific bioclimatic zones were defined

10 instead of the Köppen-Geiger climate classes used in our study. Therefore a direct comparison based on climate zones is not possible, so that a comparison by latitude remains as the most convenient approach.

In general our estimates show a much more homogeneous distribution on Earth than the estimates of the recent erosion rates. The quite inhomogeneous distribution of the latter is reflected in a strong asymmetry between the two hemispheres, a strong decrease towards the polar regions and a pronounced peak at 20°N. However, the smaller variation of our results is not

15 surprising since the climatic zones may have moved in the past as discussed in Sect. 4. our results are an average over a long time span where climate has changed and even continents have moved.

As shown in Fig. 6, the contribution of the area with an erosion rate greater than r can be approximated well by an exponential distribution  $C(r) = 0.25 \exp(-\frac{r}{200 \text{ m/Ma}})$  at high erosion rates above 250 m/Ma. This means that the area on Earth with an erosion rate greater than r decreases by about 40 % if r increases by 100 m/Ma. Qualitatively the same behavior was found

- 20 for soil losses at the plot scale, but with a decay constant about 5 times smaller (Wilkinson and McElroy, 2007). Even more striking, there is a significant deviation from the exponential decay at erosion rates below 250 m/Ma. The exponential part covers only 8% of the total ice-free land surface. This steeper decrease in the cumulative distribution at smaller erosion rates indicates that smaller areas with small erosion rates contribute much more to the total area than the exponential tail. However, when considering the contribution to the worldwide erosion, a different behavior is observed. Here, the contribution of the
- 25 large area with small erosion rates is not so high. Using our estimate of the worldwide mean erosion rate of 78 m/Ma, the data reveal that only about 25 % of the total land surface have an erosion rate above the mean, but these 25 % contribute about 75 % to total erosion. This 75 to 25 relation describes a more uneven distribution than Willenbring et al. (2014) obtained (about 70 to 30), but it is less inhomogeneous than the 80 to 20 relation often referred to as Pareto's principle in many contexts.

At this point the question may arise whether the spatial distribution of the impact craters on Earth might cause a systematic 30 error.

**6 The time scale of the terrestrial crater inventory**

As the lifetime of a crater is inversely proportional to the erosion rates, the majority of craters is found in regions with rather low erosion rates, which is confirmed by the erosion rates at the 77 craters used in the analysis shown in Fig. 6. In order to avoid a bias by the local topography of the craters, we used the at a given erosion rate depends on its size, the number of craters of different sizes should reflect the mean erosion rate of the respective province instead of the estimate at the location of the crater itself. For the temperate zone, the median erosion rates of the existing craters is 61 m/Ma. Repeating the analysis of Fig. 5 for this climate zone we found that 60of the area have a higher crosion rate , which means that 50of the craters are in these 40of

- 5 the area with the lower erosion rate, and 50in these 60of the area with a higher erosion over different time intervals. We might therefore think about an inversion approach using the crater inventory as a function of the crater size for deriving time-resolved erosion rates. This distribution seems to be not very asymmetric, but the 60of the area with a higher erosion contribute more than 92to the total erosion. As a consequence, half of the craters in the temperate zone are located in a part of the climate zone contributing less than 8to the total erosion.
- In view of this result, the estimate of Alternatively, we could use the very good fit of the worldwide crosion rate strongly relies on the assumed and to some extent verified linear relationship between relief and erosion rate. However, to our knowledge all studies in this context either found linear or slightly convex relations between morphometric parameters and erosion rates. It we assumed a convex dependency of the erosion rate on the relief, inventory assuming a constant erosion rate obtained by Hergarten and Kenkmann (2015) as evidence for a constant erosion rates over millions of years. However, the estimated
- 15 erosion rates at large relief and thus the worldwide mean erosion rate would even increase. This result would even strengthen our finding that the worldwide erosion rates on the million year scale were higher than suggested by the sedimentary record in the oceansspatial variation of erosion rates immediately tears down these such ideas. Mainly due to the variation of relief, erosion rates vary by orders of magnitude. This variation blurs the relationship between crater size and lifetime, so that no serious information about the temporal variation in erosion rates can be gathered. The obtained erosion rates remain temporal mean values, and we can only try to specify the time interval of averaging or, more precisely, the sensitivity of the mean value
- as a function of the time before present.

**7 The time scale of the terrestrial crater inventory**

According to Fig. 7, the estimated lifetimes of the considered cratersare in the range from 1 Ma to 1000 MaThe 
[revised manuscript text omitted]

We consider a domain consisting of k subdomains (here, k is between 13 and 22) of areas  $A_i$  and mean relief  $\Delta_i$ . According 15 to Eqs. 2 and 4, the expected number of craters in each subdomain is-

$$\lambda_i = \frac{A_i I}{s \Delta_i}$$

where we used the symbol  $\lambda_i$  instead of  $n_i$  in order to distinguish it from the actual number. The probability  $p_i(n_i)$  that the actual number  $n_i$  of craters occurs, is given by the Poisson distribution,

$$p_i(n_i) = \frac{\lambda_i^{n_i} e^{-\lambda_i}}{n_i!}.$$

20 Then the joint probability to find the actual combination  $n_1$ ,  $n_k$  is

$$\underline{p(n_1,\ldots,n_k)} = \prod_{i=1}^k p_i(n_i).$$

This probability depends on the parameter s via Eqs. ?? and ??. The maximum likelihood-method determines the most likely value of s in such a way that the probability to obtain the actual combination  $n_1$ ,  $n_k$  becomes maximal. For convenience, the

function-

$$\frac{L(s)}{l} \equiv \frac{\log p(n_1, \dots, n_k)}{\sum_{i=1}^k \log p_i(n_i)}$$
$$\equiv \frac{\sum_{i=1}^k (n_i \log \lambda_i - \lambda_i - \log(n_i!))}{\sum_{i=1}^k (n_i \log \lambda_i - \lambda_i - \log(n_i!))}$$

5 is maximized instead of *p* itself, so that

$$\frac{L'(s)}{\Xi} \equiv \sum_{i=1}^{k} \left(\frac{n_i}{\lambda_i} - 1\right) \frac{d\lambda_i}{ds}$$
$$\equiv \frac{-\frac{1}{s^2} \sum_{i=1}^{k} \left(n_i s - \frac{A_i I}{\Delta_i}\right)}{\frac{1}{s^2}}$$

The condition L'(s) = 0 immediately leads to Eq. 6.

**Appendix A: Confidence intervals for the estimated erosion rates**

[revised manuscript text omitted]

---

## Referee Report (RR1)

Review for esurf

Overall I feel that many of the issues raised in the original reviews were addressed, and the work has improved since the first round. Several aspects of the analysis and approach were explained much more clearly and completely, making the work both easier to follow and more convincing. In addition, there is a more extensive discussion on possible sources of error that I think is quite important.

However, I think that there still needs to be a discussion on how craters erode and how they are identified. The conclusions of the authors critically depend on the ideas that

1) The lifetime of a crater is directly related to its initial depth and the regional erosion rate.

2) The crater record is complete above a certain size

3) They have been able to correctly characterize the incompleteness of the record below that critical size

4) There is no regional bias in crater discovery.

As most terrestrial geomorphologists (the main audience of this work) are not well versed in how craters are identified on earth and how they erode on earth, it is not feasible for the average reader to critically assess how reasonable the above assumption are. A discussion of how craters erode and how this will influence their discoverability based on methods that are used to find craters would be a critical element to support the author's conclusions.

More specific points:

Section 8.1: It has been pointed out to me that there is a large body of literature on how craters erode (Craddock et al, 1997, Craddock & Howard, 2002, Forsberg–Taylor et al., 2004, Howard 2007) that is completely ignored here. Given the incredible relevance of this process to the results laid out in this paper, I think this is a critical

oversight, and while these references do not necessarily apply directly because they were developed for mars, they show both that a much more critical assessment of how craters erode is possible, and that people have already laid the groundwork for such an assessment.

Craddock, R. A., T. A. Maxwell, and A. D. Howard (1997), Crater morphometry and modification in the Sinus Sabaeus and Margaritifer Sinus regions of Mars, J. Geophys. Res., 102(E6), 13321–13340, doi:10.1029/97JE01084.

Craddock, R. A., and A. D. Howard, The case for rainfall on a warm, wet early Mars, J. Geophys. Res., 107(E11), 5111, doi:10.1029/2001JE001505, 2002.

Forsberg–Taylor, N. K., A. D. Howard, and R. A. Craddock (2004), Crater degradation in the Martian highlands: Morphometric analysis of the Sinus Sabaeus region and simulation modeling suggest fluvial processes, J. Geophys. Res., 109, E05002, doi: 10.1029/2004JE002242.

Alan D. Howard, Simulating the development of Martian highland landscapes through the interaction of impact cratering, fluvial erosion, and variable hydrologic forcing, Geomorphology, Volume 91, Issues 3-4, 2007, Pages 332-363, ISSN 0169-555X, https://doi.org/10.1016/j.geomorph.2007.04.017.

Lines 29-33: This argument only makes sense if craters are found by identification of altered rocks in the environment. I believe that this would be the case for very large craters, but how are smaller craters actually found? Are they identified initially by the rocks of the crater floor? Or are they initially found based on morphological clues that might appear to a nearby observer or in a DEM? In any case, this story also allows for a period of time where craters may be hidden by sediments deriving from their own erosion.

Section 8.2: Lines 27-29: This is cool! I very strongly think you should show the results from this analysis alongside the original results, at least for the overall erosion

rate. Seeing agreement between the two size distributions will help the overall argument because it is easier to imagine that >6km craters are tied to regional erosion rates. Also, if the error from this method is extremely large, that will motivate the use of the less dependable but larger record of smaller craters. But why have you used a different I for the larger craters? For the original analysis, you use I=5e-5 for the entire record, including these large craters. Should you not keep the same value of I then for this secondary analysis?

8.3: "However, the uncertainty arising from this relationship should be clearly smaller than the statistical uncertainty." - I'm not sure this sort of statement is useful. It sounds a bit like an opinion, rather than a quantitatively supported statement. Without any numbers for statistical uncertainty or uncertainty in I we cannot evaluate such a statement. You should either attach numbers here, or drop this statement.

Section 8.4: End of the section: "However, this is unrealistic, and we expect the potential bias to be much smaller." - This statement again sounds more like an opinion. I think you could make a stronger argument with a statement like: "while relief has surely changed, orogens are long lived, lasting for tens of millions of years (citation - Whipple?) with response times to changing boundary conditions on the scale of 1e5-1e6 years (Whipple). Therefore we expect there to be a correlation between modern and past relief on the time scale of 10 mya.

Section 8.5: This is a really great example to describe the approach taken here!

Section 8.7: "Going a step beyond the occurrence of hiatuses in the erosional history discussed in the previous section, intermittent phases of sedimentation should also be taken into account as a potential source of errors" - phases of sedimentation are generally considered to be one of the main mechanisms for a hiatus in erosion.

Line 28-29: True, it faithfully records the longterm average rate of erosion, but it breaks the relationship between erosion rate and boundary conditions, so it would mess up any correlation between say climate or relief and longterm erosion rates.

Section 8.7: I really appreciate the attempt throughout this entire section 8 to account for error in the approach. However, I don't agree with the last paragraph at all. It seems to me that you strongly underplay the effect of missing craters in the record. This is also essentially a one way bias that only leads to an overestimation of erosion rates. Unless it is possible to identify something else as a crater by accident, craters can only be missed. Even the argument in section 8.7 allows for the idea that craters can be buried until a given date, which may not be today. It seems prudent to allow for the possibility that the incompleteness has been mischaracterized (given the few craters that available to characterize it) similarly it seems prudent to allow for a subset of craters that were eroded or erased much faster than the background average erosion rates would imply. I think to try to sum the sources of error and imply that you could have only underestimated erosion rates is coy. Error is error. In such an approach with so many unknowns over such a long period of time, I would not end with such a statement as you do here.

---

## Author Response (AR2)

**Notes**

- Referee comments are marked with green and red color (critical comments).

- Responses are displayed in black.

- References to changes in the manuscript are marked in blue color; line numbers refer to the version with marked changes (latexdiff).

**Reviewer 1**

Overall I feel that many of the issues raised in the original reviews were addressed, and the work has improved since the first round. Several aspects of the analysis and approach were explained much more clearly and completely, making the work both easier to follow and more convincing. In addition, there is a more extensive discussion on possible sources of error that I think is quite important.

However, I think that there still needs to be a discussion on how craters erode and how they are identified. The conclusions of the authors critically depend on the ideas that

1) The lifetime of a crater is directly related to its initial depth and the regional erosion rate.

2) The crater record is complete above a certain size

3) They have been able to correctly characterize the incompleteness of the record below that critical size

4) There is no regional bias in crater discovery.

As most terrestrial geomorphologists (the main audience of this work) are not well versed in how craters are identified on earth and how they erode on earth, it is not feasible for the average reader to critically assess how reasonable the above assumption are. A discussion of how craters erode and how this will influence their discoverability based on methods that are used to find craters would be a critical element to support the author's conclusions.

Ok, it may make sense to discuss the erosion of craters a bit more in detail, although I thought it was already addressed in the first revision.

Added paragraphs on page 3, lines 4–9 and page 11, lines 3–14.

**Reviewer 3**

The authors introduce a new method into the search for robust records and measurements of global erosion rate. They use the crater preservation record and derive the equations relevant to convert them into erosion rate. This is novel, exciting, and a refreshing addition to Geomorphology. This material no doubt should be published.

I notice that in a previous round of reviews the paper has been scrutinised extensively and detailed by other reviewers, and the authors have corresponded to most of the issues raised. Thus I assume the underpinnings of this paper our sound, the uncertainties and statistics have been well addressed, and I do not re-evaluate these here.

My concern is about the style of the presentation. The paper has been written by scientists of which one is more on the mathematical side of Geomorphology, and the other is an impact expert. While this combination adds the special flavour to the paper it makes the piece very hard to follow. In presenting the mathematical descriptions it is assumed that the knowledge of

these concepts by the presumably mostly geomorphological readers is there. I doubt that this is the case.

I also doubt that this is the case, but in principle this applies to all methods used in this field. As an example: How many geomorphologists were presumably able to understand the inversion procedure described in the 2013 Nature paper of Herman et al., and how many could really assess whether there could be problems? I do not think that it is possible to pick up all readers with a reasonable effort and length of the paper, and I am not convinced that the points raised below would really fix this problem. In turn, I think that those readers who read the paper completely will have a good chance not to get stuck at these points.

Examples are mainly poorly defined terms like "consumption of craters", "harmonic mean rate", "local sedimentation rates in craters" . . .

Ok, it makes sense to explain the arguments how craters vanish by sediment deposition a bit more in detail, but it seems not be very useful to explain terms such as harmonic mean.

Added some explanation (page 4, lines 14–17).

. . . and the uncertainty estimations where it is unclear whether the discussion is about true "uncertainty" or "error" (of measurements).

As all erosion rates are maximum likelihood estimates, it should be clear in which way the uncertainties in the given numbers should be interpreted.

Most of this is easy to fix by some clear definitions. Similarly, the geomorphological interpretation suffers from a mix of awkward views, outdated comparison concepts, and terms that are neither defined nor seem to be used rigorously. Examples are the use of relief as prime driver which seems outdated where today slope, or even more so, river steepness and hence tectonic uplift have been identified as prime erosion drivers, . . .

To my knowledge, this has never been evaluated systematically since the study of Summerfield and Hulton (1994). Average slope has indeed been used in more recent studies, but the Geology paper of Willenbring et al. (2013) is a good example for the pitfalls and illustrates that the advantage of slope over relief is in sum not clear. And I agree that quantitative tectonic geomorphology mainly relies on channel slopes and the respective normalized steepness indices as they allow a between spatial resolution and reflect process-based ideas about fluvial erosion. However, there are many nontrivial issues such as the question for which part of a catchment a segment of the trunk channel can be seen as representative. In my opinion there is no reason against using Ahnert's old concept of relief. Apart from this river steepness is by no means more closely related to tectonic uplift than the other metrics at large scales.

Added the usage of channel slopes and steepness index (page 2, lines 10 and 14–15) and some more explanation why relief is used (page 4, line 21 to page 5, line 2).

. . . the use of the Köppen climate categories where I know of no process-based prediction why these should be in any way relate to erosion, . . .

I do not know of any process-based prediction why they should relate to erosion, but those readers going through the manuscript completely will find the explanation why this does not matter at all.

. . . "efficiacy" of erosion, . . .

This term was clearly defined (page 6, line 14).

. . . "erosive regimes".

I cannot imagine that any reader might have problems with this term even without a formal

definition after this term was used the first time on page 5, lines 16–21.

In addition, I found a few items a bit inconsistent with much of the newer literature.

- The assumption that cooling climate should lead to lower erosion rates (most work assumes it is the opposite, an increase mainly arising from the strong cold/warm period swings and sealevel change associated with the Quaternary)

  It is only stated stated that cooling should result in lower weathering rates and thus in lower erosion rates. Undoubtedly some ideas beyond glaciation have been proposed why erosion could increase even in cooling periods (those mentioned above), but it is clear that they have been developed in order to explain the counterintuitive increase of erosion. As our results do not confirm this increase, it does not make much sense to explain these concepts here.

- That glaciation is a big driver of erosion (probably true for mountains, but not for the northern or antarctic glacial sheets)

  It is only stated that it is in discussion as a big driver, and this is definitely true.

- That warm climate facilitates weathering (correct as long as it is about weathering degree not rate) but that erosion should simultaneously be high is incorrect unless there is a tectonic driver

  This was never claimed in the manuscript.

- Fundamentally, the attempt to discern the driver that sets erosion rate is too much for this paper – many other approaches with much higher spatial resolution (like cosmogenic nuclides and river loads) have failed to do so as this is such a multivariate problem. As such, I fail to grasp the significance of e.g. figures 1 and 3.

  I think that those readers who read the paper completely recognize that the erosional efficacies are lumped values and do not represent the actual climatic conditions exactly, but in turn avoid the problem of multivariate dependencies present in other methods where point data must be extrapolated. So the spatial resolution is not an argument at all here.

  Added a remark about the lumped values in the conclusions (page 15, lines 15–18).

- It is stated that the ocean 10Be/9Be is recording erosion rates (it doesn't, the proxy records weathering rate and by inference denudation (erosion + weathering) rate

  The use of the terms "erosion" and "denudation" is not unique in the literature since Gilbert and Penck. As long as even Portenga and Bierman use "erosion" in their review on 10Be, we prefer to use this term throughout the manuscript, too. It should be clear to the readers that it is just how much material is removed from the surface.

- And I wonder whether the time integration of craters over millions of years makes a two-fold increase in erosion, that newer literature assumes as maximum increase (Herman et al.) in the late Cenozoic, even detectable within the uncertainties of the method.

  It is written at the beginning of Sect. 6 that the method itself would only be able to detect temporal changes in erosion rates if the rates were spatially homogeneous. However, for those readers who only read the introduction and the conclusion we have now added a statement in the conclusions that all about changes in erosion rates has been derived by comparing absolute values, and that there is no way to detect changes directly with out approach.

  Added a remark in the conclusions (page 15, lines 31–34).

- In that context it is questionable whether the objective stated in the title "Has erosion rate globally increased" can be answered – better drop that first part of the title.

    Even if the question cannot be answered completely, I do not see any argument against raising the question in the title.

Nothing of these statements makes this paper wrong, but just difficult to read, which will, possibly, lead to a lack of acceptance by the Geomorphology community which would be a shame given the innovative approach. What I suggest is thus either of two rather bold editorial change options.

1) Limit this paper to providing the reader with a didactic, careful, and step-by step explanation of the way the used metrics are developed, including, perhaps, some cartoons on, for example, crater erosion and crater filling by sediment. And drop all detailed geomorphic motivation / conceptual interpretation and reduce these to a short summary paragraph at the end where results are compared with other estimates. It would essentially be a methods paper, with profound results though.

2) Take another geomorphological co-author on board who is familiar with the up-to-date view of global erosion, its driving forces, their changes in the past, and the data that is there for comparison, with the aim to present that comparison with estimates at the state of the art.

Either option will make this a great paper. Reluctantly, I could also accept publishing this paper with its present contents (which is not the best option), but in that case the lingual style needs some much editing.

I partly agree concerning the acceptance of the paper in the community of geomorphology. But on the other hand, I am not convinced that the suggested options would really solve this problem. There are two opposite opinions concerning the potential increase in erosion rates, and the paper will be one more contribution supporting the (probably at this time smaller) side finding no increase. But no matter how much effort we spent in making it smoother for the geomorphology community, it will not be the final point in this controversial discussion.

[revised manuscript text omitted]

---

## Author Response (AR3)

Dear Jean,

thanks for the editorial handling. I adjusted the title, the sentence concerning the weathering rates and made a few small fixes.

All the best,

*Stefan*

Albert-Ludwigs-Universität
Freiburg

Institut für Geo- und
Umweltnaturwissenschaften

Abteilung Geologie

Prof. Dr. Stefan Hergarten
Professor für Oberflächennahe
  Geophysik

Albertstraße 23 b
79085 Freiburg

Tel. 0761/203-6471
Fax 0761/203-6496

stefan.hergarten@
  geologie.uni-freiburg.de
www.hergarten.at

Freiburg, 18. 4. 2019

[revised manuscript text omitted]